# Estimates of spatially complete, observational data-driven planetary boundary layer height over the contiguous United States

Zolal Ayazpour[1], Shiqi Tao[2,a], Dan Li[2], Amy Jo Scarino[3], Ralph E. Kuehn[4], and Kang Sun[1]

[1]Department of Civil, Structural and Environmental Engineering, University at Buffalo, Buffalo, NY, USA
[2]Department of Earth and Environmental Sciences, Boston University, Boston, MA, USA
[3]NASA Langley Research Center, Hampton, VA, USA
[4]Space Sciences Engineering Center, University of Wisconsin-Madison, Madison, WI, USA
[a]now at Department of Geography, Clark University, Worcester, MA, USA

**Correspondence:** Kang Sun (kangsun@buffalo.edu)

**Abstract.** This study aims to generate a spatially complete planetary boundary layer height (PBLH) product over the contiguous US (CONUS). An eXtreme Gradient Boosting (XGB) regression model was developed using selected meteorological and geographical data fields as explanatory variables to fit the PBLH values derived from Aircraft Meteorological DAta Reporting (AMDAR) hourly profiles at 13:00–14:00 local solar time (LST) during 2005–2019. The PBLH prediction by this work as well as PBLHs from three reanalysis datasets (ERA5, MERRA-2, and NARR) were compared to reference PBLH observations from spaceborne lidar (CALIPSO), airborne lidar (High Spectral Resolution Lidar, HSRL), and in-situ research aircraft profiles from the DISCOVER-AQ campaigns. Compared with PBLHs from reanalysis products, the PBLH prediction from this work shows closer agreement with the reference observations, with the caveat that different PBLH products and estimates have different ways of identifying the PBLH and thus their comparisons should be interpreted with caution. The reanalysis products show significant high biases in the west CONUS relative to the reference observations. One direct application of the dataset generated by this work is that it enables sampling PBLH at the sounding locations and times of sensors aboard satellites with overpass time in the early afternoon, e.g., the A-train, Suomi-NPP, JPSS, and Sentinel 5-Precursor satellite sensors. Since both AMDAR and ERA5 are continuous at hourly resolution, the observational data-driven PBLHs may be extended to other daytime hours.

## 1 Introduction

The planetary boundary layer (PBL) is the lowest part of the atmosphere that mediates the exchange of momentum, energy, and mass between the surface and the overlying free troposphere (Stull, 1988). It plays a central role in land-atmosphere coupling, linking surface states and characteristics (e.g., surface temperature, soil moisture, and vegetation) to convection through surface fluxes of sensible and latent heat (Santanello et al., 2018). Improving our understanding and characterization of the PBL is critical for enhancing the predictability of numerical weather prediction and global climate and earth system models (Garratt, 1994; Stensrud, 2009). The PBL height (PBLH), which characterizes the vertical extent of the PBL, is a critical parameter in many land-atmosphere coupling metrics (Santanello et al., 2013, 2015). The PBLH also governs the vertical mixing of thermal energy, water, and trace gases and hence strongly regulates the near-surface pollutant concentrations. With the same amount

of emission, a higher PBLH typically means stronger dilution and hence lower near-surface concentrations, and vice versa for a lower PBLH. Therefore, accurate knowledge of the PBLH is important in the modeling of air quality through a chemical transport model (Zhu et al., 2016) and in the inference of emissions through inverse methods (Gerbig et al., 2008). The PBLH can also help bridge the gaps between the column-integrated quantities observed by satellites and near-surface concentrations for short-lived species such as fine aerosols (Su et al., 2018), $NO_2$ (Boersma et al., 2009), and $NH_3$ (Sun et al., 2015). In order to sample the PBLH values at the satellite sounding locations and times (Sun et al., 2018), extensive spatiotemporal coverage of the PBLH data is needed.

Spatially and temporally complete estimates of the PBLH can be provided by atmospheric models or reanalysis products, although the PBLH is in general not directly simulated, but diagnosed from model profiles of wind, temperature, or turbulent kinetic energy (TKE). Besides the inherent modeling uncertainties, the PBLH-estimating methods and the associated empirical parameters in these methods introduce additional uncertainties and inconsistency among models. The PBLH products from three commonly used reanalysis products are evaluated in this work, including the fifth generation European Centre for Medium-Range Weather Forecasts (ECMWF) atmospheric reanalyses of the global climate (ERA5) (Hersbach et al., 2020), the Modern Era Retrospective analysis for Research and Applications version 2 (MERRA-2) from NASA (Gelaro et al., 2017), and the North American Regional Reanalysis (NARR) from NOAA National Centers for Environmental Prediction (NCEP) (Mesinger et al., 2006). The PBLH values were determined by different methods in these three reanalysis products: critical bulk Richardson number for ERA5, threshold total eddy diffusion coefficient of heat for MERRA-2, and threshold TKE value for NARR. Validations of those model-based PBLH products by observations are limited, and discrepancies are frequently observed (Zhang et al., 2020). The PBLHs from MERRA-2 and NARR are widely used in GEOS-Chem and WRF-Chem chemical transport models, respectively (Lu et al., 2021; Murray et al., 2021; Laughner et al., 2019; Hegarty et al., 2018). If the PBLHs from those reanalysis models were biased, the associated chemical transport model simulations would be directly impacted. McGrath-Spangler and Denning (2012) found that the PBLH in MERRA (the previous version of MERRA-2) was higher than retrieved PBLH from spaceborne lidar CALIOP (Cloud-Aerosol Lidar with Orthogonal Polarization) aboard the Cloud-Aerosol Lidar and Infrared Pathfinder Satellite Observation (CALIPSO) satellite over the arid and semiarid regions of North America, where NARR gave even higher PBLH values than MERRA. Comparisons between PBLH from GEOS-FP, the assimilated meteorological fields generated by the same model used for MERRA-2 (i.e., the Goddard Earth Observing System, Version 5, GEOS-5 model) and lidar/ceilometer data showed a 30–50% high bias in the southeast US (Zhu et al., 2016; Millet et al., 2015). Zhang et al. (2020) compared the ERA5 PBLHs with those derived from hourly profiles measured by commercial airlines in the Aircraft Meteorological DAta Reporting (AMDAR) (Zhang et al., 2019) and found that ERA5 overestimates the daytime PBLH over the contiguous US (CONUS) by 18–41%. Furthermore, models and reanalysis products often disagree with continuous observations in terms of the diurnal evolution of PBLH (Zhang et al., 2020; Hegarty et al., 2018).

In spite of its importance, observations of PBLH are sparse in both space and time. Operational radiosondes launched twice-daily at 0000 and 1200 UTC have been used to construct long-term PBLH climatology (Guo et al., 2016; Seidel et al., 2012), but the launching times largely missed the daytime variations of PBLH over the CONUS (Zhang et al., 2020). On the other hand, intensive in-situ atmospheric profiling by radiosondes or aircraft spiraling profiles (e.g., the NASA Deriving Information

on Surface Conditions from Column and Vertically Resolved Observations Relevant to Air Quality, or DISCOVER-AQ campaign) are limited to short periods and certain locations (Angevine et al., 2012; Zhang et al., 2016). In addition to radiosondes and aircraft spiraling profiles, ground-based, airborne, and space-borne lidar remote sensing of aerosol backscatter also provide PBLH observations. For example, Hegarty et al. (2018) synergized a network of ground-based micropulse lidar (MPL), an airborne High Spectral Resolution Lidar (HSRL), and space-borne lidar CALIPSO to examine the diurnal PBLH variations during the DISCOVER-AQ campaign in Maryland-DC. Compared to surface and suborbital lidar observations, CALIPSO provides better spatial (global coverage) and temporal (from 2006 to present) coverage. However, there are no official CALIPSO PBLH product available, and existing PBLH retrievals from CALIPSO Level 1B data have low temporal resolution as the CALIPSO ground track is sparse with 16-day repeat cycle (Jordan et al., 2010; McGrath-Spangler and Denning, 2012, 2013; Su et al., 2018). The other PBLH observation data source is AMDAR which provides global automated weather reports from commercial aircrafts (Moninger et al., 2003). Zhang et al. (2020) derived PBLH from AMDAR profiles and constructed a continuous hourly climatology of PBLH at 54 major airports in the CONUS. The same methodology developed by Zhang et al. (2020) can be readily applied to other regions in the world where major commercial airports exist and other time periods given the operational status of AMDAR observations. Compared to the existing CALIPSO PBLH product, the hourly PBLH derived from AMDAR profiles features a more complete temporal coverage and higher signal-to-noise ratios. Therefore, the AMDAR PBLH dataset is selected as the observational PBLH data source for this research.

The existing AMDAR PBLH dataset is available hourly from 2005 to 2019 at 54 airport locations. However, many applications require PBLH in other locations or completely covering a region. The objective of this study is to produce observation-based, spatially-complete PBLH fields over the CONUS. We develop a data-driven predictive model using various meteorological and geographical predictors to match the AMDAR PBLH observations. Since the predictors all have complete spatial coverage over the CONUS, running the model forwardly can yield PBLH prediction at arbitrary locations in the domain. We cross-validate the model in space by randomly splitting the airports into training and testing sets, and the model is selected based on averaged metrics on the testing sets. The predicted PBLH is then compared to PBLHs from three widely-used reanalysis products (ERA5, MERRA-2, and NARR), and all of them are further compared to independently diagnosed PBLH from observations (e.g., from research aircraft profiles and HSRL airborne lidar) and the CALIPSO PBLH product. Here we should emphasize that such comparisons should be interpreted with caution since different PBLH products and estimates have different ways of identifying the PBLH. In this sense, none of the PBLH products and estimates should be treated the as golden truth. However, such comparisons remain meaningful because (1) they provide confidence in the PBLH prediction by this work and (2) they generate information regarding the difference between various PBLH products and estimates that have been widely used in previous work.

One direct application of the PBLH product from this work is that it enables sampling PBLH at the sounding locations and times of satellite sensors with overpass time in the early afternoon, e.g., the A-train sensors as well as sensors on board of similar polar-orbiting platforms, such as CrIS and TROPOMI. Since both AMDAR and ERA5 are continuous at hourly resolution, future work may extend to other daytime hours observable by geostationary missions like TEMPO (Zoogman et al., 2017) and potentially to existing satellites with drifted orbits such as the NASA Earth Observing System (EOS) satellites.

## 2 Data

The training data used to estimate spatially complete PBLH over the CONUS were obtained from AMDAR observations. The relevant meteorological data from ERA5 as well as geographical data were used as explanatory variables, i.e., predictors or features, in a data-driven predictive model to predict PBLH at AMDAR observation locations. The model prediction at locations other than AMDAR observation locations was evaluated using three independent observational data sets, including (1) the Surface Attached Aerosol Layer product from CALIPSO, (2) NASA HSRL observations during the DISCOVER-AQ campaigns and the Studies of Emissions, Atmospheric Composition, Clouds and Climate Coupling by Regional Surveys (SEAC4RS) campaign, and (3) manually labeled PBLH from spiraling profiles during DISCOVER-AQ. In this study, we only focus on data at 1300–1400 LST. In addition to the validation with ground measurements, we also compared our estimations with PBLH data from ERA5, MERRA-2, and NARR. Detailed descriptions of these data sets are provided as follows.

### 2.1 The AMDAR observations

AMDAR provides meteorological data captured by sensors aboard commercial aircrafts worldwide. In this study, we used all quality assured data from 54 airports within the CONUS from 2005 to 2019. The locations of these airports are shown in Fig. 1a. Vertical profiles of potential temperature, humidity, and wind speed were averaged over each UTC hour at each airport (Zhang et al., 2019). The PBLH was derived using the bulk Richardson number method (Seidel et al., 2012; Zhang et al., 2020), where the bulk Richardson number is calculated as

$$Ri_b = \frac{(g/\theta_{v,s})(\theta_{v,z} - \theta_{v,s})(z - z_s)}{(u_z - u_s)^2 + (v_z - v_s)^2 + bu^{*2}}. \tag{1}$$

Here $g$ is gravity acceleration, $\theta_v$ is virtual potential temperature, $u$ and $v$ are horizontal winds, and $u^*$ is the surface friction velocity. Subscript $z$ denotes vertical profile values and subscript $s$ denotes value at the lower boundary of PBL. We use $z_s = 40$ m, $b = 100$, and critical bulk Richardson number of $0.5$ following Zhang et al. (2020). Regarding $u^*$, Zhang et al. (2020) tested two options, one using $u^*$ from ERA5 and the other one using a constant $u^* = 0.3$ m s$^{-1}$, and found the results were similar. We choose the latter option so that the AMDAR PBLH is purely observation-based and independent of the reanalysis.

Although the bulk Richardson number method was found to be the most robust one to derive PBLH from AMDAR measured profiles, significant challenges remain under stable conditions. Figure 2a-c compare all AMDAR-derived PBLH at 1300–1400 LST with spatiotemporally interpolated PBLH from ERA5. Under stable and, to a lesser extent, neutral conditions, a significant portion of data cluster close to the horizontal axis (panels a and b), indicating that AMDAR gives very low PBLH ($< 500$ m) whereas ERA5 reports a wide range of PBLH values up to 3000 m. We did not find any meteorological or geographical factors that would explain the occurrence of AMDAR vs. ERA5 PBLH data pairs in these clusters. The most likely cause of these clusters of data is the ambiguity of identifying the PBL top. Under challenging conditions, the critical bulk Richardson number algorithm that is used in AMDAR and ERA5 data to diagnose PBLH may identify different vertical structures as PBL top, leading to very different and uncorrelated PBLH values. Including these uncorrelated clusters of points will strongly bias the model training results. Consequently, we use quantile regression (Hirschi et al., 2011) to filter them out to avoid significant bias in the model training.

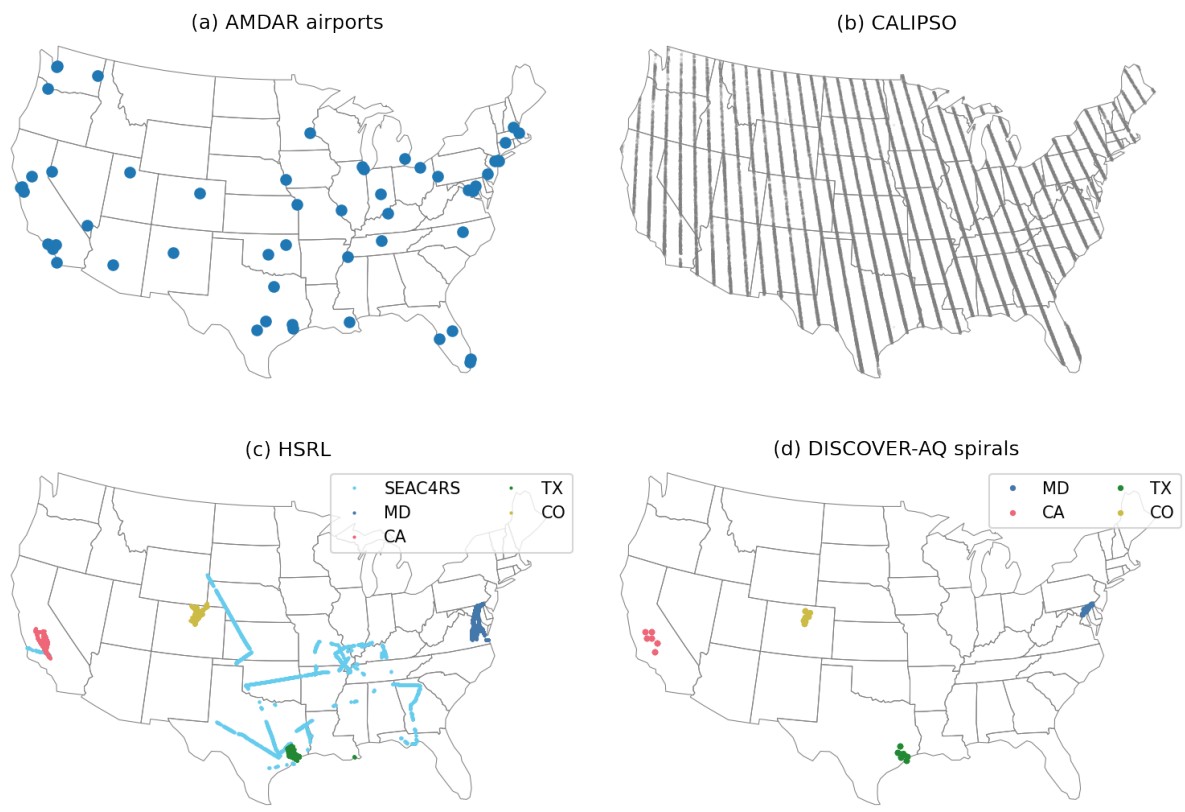

**Figure 1.** Overview of the locations of the main source of PBLH observation dataset AMDAR (a) and the three independent validation datasets from spaceborne LIDAR CALIPSO (b), airborne LIDAR HSRL (c), and in-situ aircraft spiral profiling during the DISCOVER-AQ campaigns (d). The HSRL and spiral profile data were limited to 1230–1430 LST only, and the CALIPSO overpasses were always at around 1330 LST.

Quantile regression fits a series of regression lines that correspond to different quantiles of the AMDAR PBLH distribution conditioned on the ERA5 PBLH values. The distribution of slopes from 0 to 100 quantile lines is displayed in Fig. 2d. The overall slopes are smaller than unity, indicating AMDAR PBLH is generally lower than ERA5. The slopes show a bimodal distribution due to the two clusters of AMDAR-ERA5 relationship. We choose a slope threshold of 0.5 (highlighted in Fig. 2d) to separate the data cluster near the 1:1 line and the data cluster near the horizontal axis. The quantile regression line corresponding to a slope of 0.5 is labeled and plotted in Fig. 2a-c. All data points below this line were removed from further analysis, which accounted for about one third of all AMDAR data, half of data under stable condition, and only $10\%$ of data under convective condition. The distribution of AMDAR PBLH after such a filtering is shown in Fig. 3a.

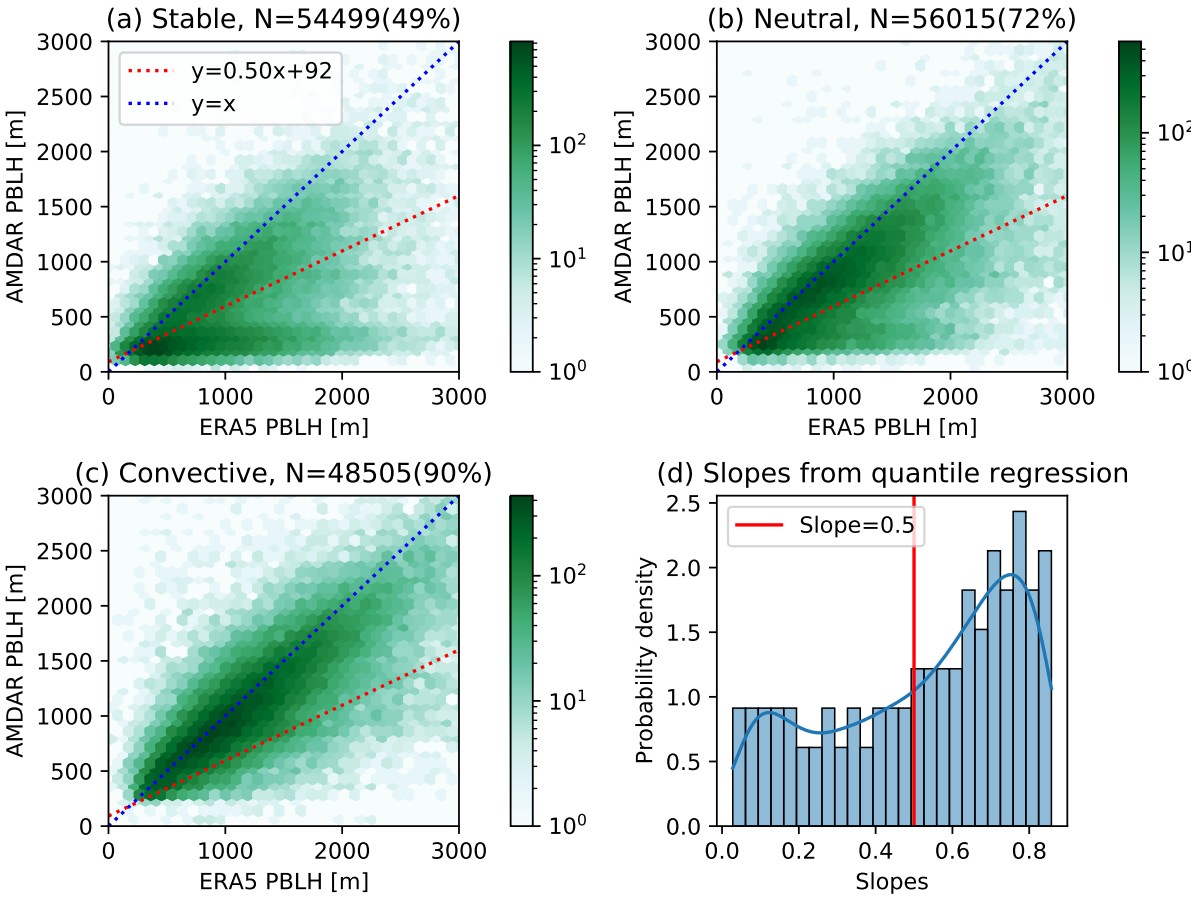

**Figure 2.** Comparison between PBLHs from AMDAR and ERA5 under stable (a), neutral (b), and convective (c) conditions, where atmospheric stability is classified using AMDAR profiles following Zhang et al. (2020). The blue and red dashed lines are the 1:1 line and the quantile regression line corresponding to a slope of 0.5, respectively. The titles give the total number of points as well as the percentage of data points above the 0.5-slope line, i.e., the fraction of data after filtering. (d) shows the distribution of quantile regression slopes, where 0.5 marks the threshold between two clusters.

## 2.2 Reanalysis datasets

### 2.2.1 ERA5

ERA5 provides multiple climate variables at a spatial resolution of 0.25 degree (approximately 30 km) for the globe every hour, with 137 levels from the surface up to 0.01 hPa (around 80 km height). Currently ERA5 extends from 1970 to 5 days earlier than the present date. Hourly single-level ERA5 fields (ECMWF, 2020) were obtained from the Copernicus Climate Data Store (https://cds.climate.copernicus.eu/) and spatiotemporally sampled at the coordinate and time of AMDAR observations as explanatory variables for model construction. The ERA5 PBLH was also spatiotemporally sampled at other independent observation coordinate and time for intercomparison. The PBLH from the ERA5 product is identified using the bulk Richardson number method but with slightly different parameters (ECMWF, 2017). Zhang et al. (2020) compared AMDAR PBLH estimated with the same parameters as ERA5, but the results gave larger biases relative to ERA5 and other observations.

### 2.2.2 MERRA-2

Extending from 1980 to around one month earlier than the present date, MERRA-2 is the first long term global reanalysis data that assimilates observational records of aerosol and its impact on other physical processes. MERRA-2 is widely used as meteorological fields for the GEOS-Chem chemical transport model (Hu et al., 2017; Lu et al., 2021; Murray et al., 2021). The spatial resolution of MERRA-2 is $0.5 \times 0.625°$. The PBL top pressure in MERRA-2 is identified as the model level under which the total eddy diffusion coefficient of heat falls below the threshold value of 2 $m^2$ $s^{-1}$ (McGrath-Spangler and Molod, 2014). We obtained hourly PBL top pressure and surface pressure fields from MERRA-2's time-averaged single-level diagnostics collection (GMAO, 2015) and calculated PBLH using a scale height of 7500 m. The MERRA-2 PBLH was then spatiotemporally sampled at the independent observation coordinate and time for intercomparison.

### 2.2.3 NARR

The NARR reanalysis covers the North America with a 3-hour temporal resolution and 32-km spatial resolution. It is commonly used as initial and boundary conditions to drive WRF and WRF-Chem simulations for atmospheric composition studies (e.g., Laughner et al., 2019; McKain et al., 2012; Hegarty et al., 2018). The NARR PBLH is computed from the TKE profile which is computed with a level-2.5 Mellor–Yamada closure scheme (Lee and De Wekker, 2016). We obtained the NARR PBLH field from the NCAR Research Data Archive (NCEP, 2005) and sampled it at observational coordinate/time for intercomparison.

## 2.3 Observational datasets used for evaluation

### 2.3.1 CALIPSO

The CALIPSO satellite was launched into the A-train in 2006, carrying the CALIOP instrument, a two-wavelength (532 and 1064 nm) polarization sensitive lidar. The CALIPSO observation is only available at nadir and around 13:30 LST with a 16-day repeat cycle. A wavelet covariance transform analysis technique similar to Davis et al. (2000) was applied to

identify the surface-attached aerosol layer height as a proxy of PBLH. The algorithm has been improved to work with the lower signal-to-noise ratio (SNR) data provided by CALIPSO and automated to generate global data without intervention. The retrieval methodology was previously validated through comparison with Tropospheric Airborne Meteorological Data Reporting (TAMDAR) observations and HSRL observations from aircraft platforms. We obtained the CALIPSO data from 2006 to 2013 from the University of Wisconsin-Madison Space Science and Engineering Center website (https://download.ssec.wisc.edu/files/calipso/). The spatial distribution of CALIPSO sampling locations is shown in Fig. 1(b). During the evaluation, we first obtain the model prediction at the same location and time of each CALIPSO sounding and then compare the predicted PBLH with the CALIPSO PBLH. The CALIPSO data provide much more uniform spatial and temporal coverage than the other validation datasets from airborne campaign, which are only available in limited spatiotemporal ranges. One should note that the backscatter measurements from air- and space-borne lidar characterize the mixed layer or surface-attached aerosol layer height, which may be deeper than the PBLH derived based on temperature profiles. In this study, we consider all these as different estimates of PBLH. The distribution of all CALIPSO PBLH is also plotted in Fig. 3a. The CALIPSO PBLH distribution (mean value of 1807 m) is significantly shifted to higher values relative to the distribution of AMDAR PBLH (mean value of 1047 m), with the caveat that the distributions of CALIPSO and AMDAR are not directly comparable given their different sampling and the large uncertainty from CALIPSO.

### 2.3.2 The NASA HSRL observations

We compiled NASA Langley Research Center airborne HSRL observations from DISCOVER-AQ campaigns in Maryland-DC (simplied as MD, July 2011), California (CA, January–February 2013), Texas (TX, September 2013), and Colorado (CO, July–August 2014) and the SEAC4RS campaign (carried out by the DIAL/HSRL instrument) over the southeast US in August–September 2013. A Haar wavelet algorithm was applied to automatically identify the sharp gradients in aerosol backscatter located at the top of the PBL, where the aerosol backscatter profiles were computed every 0.5 s using a 10 s running average of the HSRL 532 nm backscatter data (Hair et al., 2008; Scarino et al., 2014). We used the "best estimate" field from the HSRL data, which was a combination of the automated Haar wavelet algorithm product and visually inspection of the backscatter image. One-minute running mean averages of the "best estimate" HSRL PBLH were used in the intercomparison with other PBLH estimates to further reduce random errors. The HSRL data are restricted between 1230 and 1430 LST and the sampling locations are displayed in Fig. 1c. The distributions of HSRL PBLH in the five campaigns are shown in the right column of Fig. 3.

### 2.3.3 DISCOVER-AQ spiral profiles

During the DISCOVER-AQ campaigns, the NASA P-3B aircraft which was equipped with a suite of in-situ atmospheric sensors systematically sampled the PBL and lower free troposphere, covering wide ranges of pollution levels, atmospheric stability, and local times. The horizontal heterogeneities that confound the interpretation of conventional airborne vertical measurements were greatly reduced by spiral profiling. "Missed approach" maneuvers were also performed near some spiral sites to extend profiles to as low as 25 m above the ground. The PBLH was manually determined from vertical profiles of

potential temperature, relative humidity, aerosol extinction, and the mixing ratios of water vapor, $CO_2$, and $NO_2$ from the merged 1-Hz data. More details of this manually labeled PBLH dataset can be found in Zhang et al. (2020). The spiral profiles concentrate at a few sites that are highlighted for each phase of the campaign in Fig. 1d. Similar to the HSRL data, we also restricted the spiral profile-based PBLH between 1230 and 1430 LST. The distributions of PBLH in the four DISCOVER-AQ campaigns are shown in Fig. 3b-e. Note the number of spiral profile-based PBLH data is much lower than HSRL-based PBLH, although their distributions in the same campaigns are consistent.

### 2.3.4 Comparisons of observational datasets

As summarized by Figs. 1 and 3, none of the observational datasets described above can uniformly represent the PBLH over the study domain. CALIPSO features the most homogeneous spatial coverage (Fig. 1b), but its PBLH product relies on an automatic, global algorithm that may be subject to significant uncertainties. Yet the unique benefit of including CALIPSO data is that it can indicate errors in the spatial prediction made by our model, as the availability of AMDAR airports is spatially clustered (Fig. 1a). For example, no AMDAR sites are available in the large area over the Northern Rockies and Plains and the Southeast. Because of the large differences in AMDAR and CALIPSO PBLH, we consider the intercomparison involving CALIPSO more relative than absolute and focus on correlations rather than biases.

One should also note that CALIPSO and HSRL PBLH data are based on aerosol backscatter gradients, which is quite distinct from AMDAR, DISCOVER-AQ spiral profiles, and ERA5, where PBLH values are diagnosed thermodynamically. Although systematic differences between aerosol-based and thermodynamics-based PBLH may exist, we do not observe them by comparing spatiotemporally close spiral and HSRL measurements in the same DISCOVER-AQ campaigns (i.e., comparing d vs. g, c vs. h, d vs. i, and e vs. j in Fig. 3). Furthermore, the model prediction from this work may serve as a "traveling standard" when evaluated against HSRL and spiral datasets. As will be shown in Sections 4.2 and 4.3, the biases between HSRL data and collocated model prediction do not show significant differences from the biases between spiral data and the corresponding model prediction.

## 3 PBLH modeling and prediction

In this study, a data-driven machine learning approach is applied to produce a spatially-complete PBLH dataset. In recent years, atmospheric scientists have applied different machine learning approaches to successfully address various atmospheric problems. For example, Jiang and Zhao (2022) applied a machine learning algorithm to calculate the PBLH from the GPS radio occultation data and investigated the seasonal variation of PBLH at multiple stations. Here, we also use a machine learning approach to estimate the PBLH at the locations where no AMDAR PBLH data are available in the CONUS.

The observational dataset consists of AMDAR PBLH at 1300-1400 LST from 2005 to 2019 after filtering are shown in Fig. 2. Predictors were sampled spatially and/or temporally from ERA5 and geographical datasets. A data-driven, non-parametric model is trained to minimize the difference between model prediction and observational data, as indicated by Eq. 2

$$\mathbf{y} = \mathbf{f}(\mathbf{X}) + \mathbf{e}, \tag{2}$$

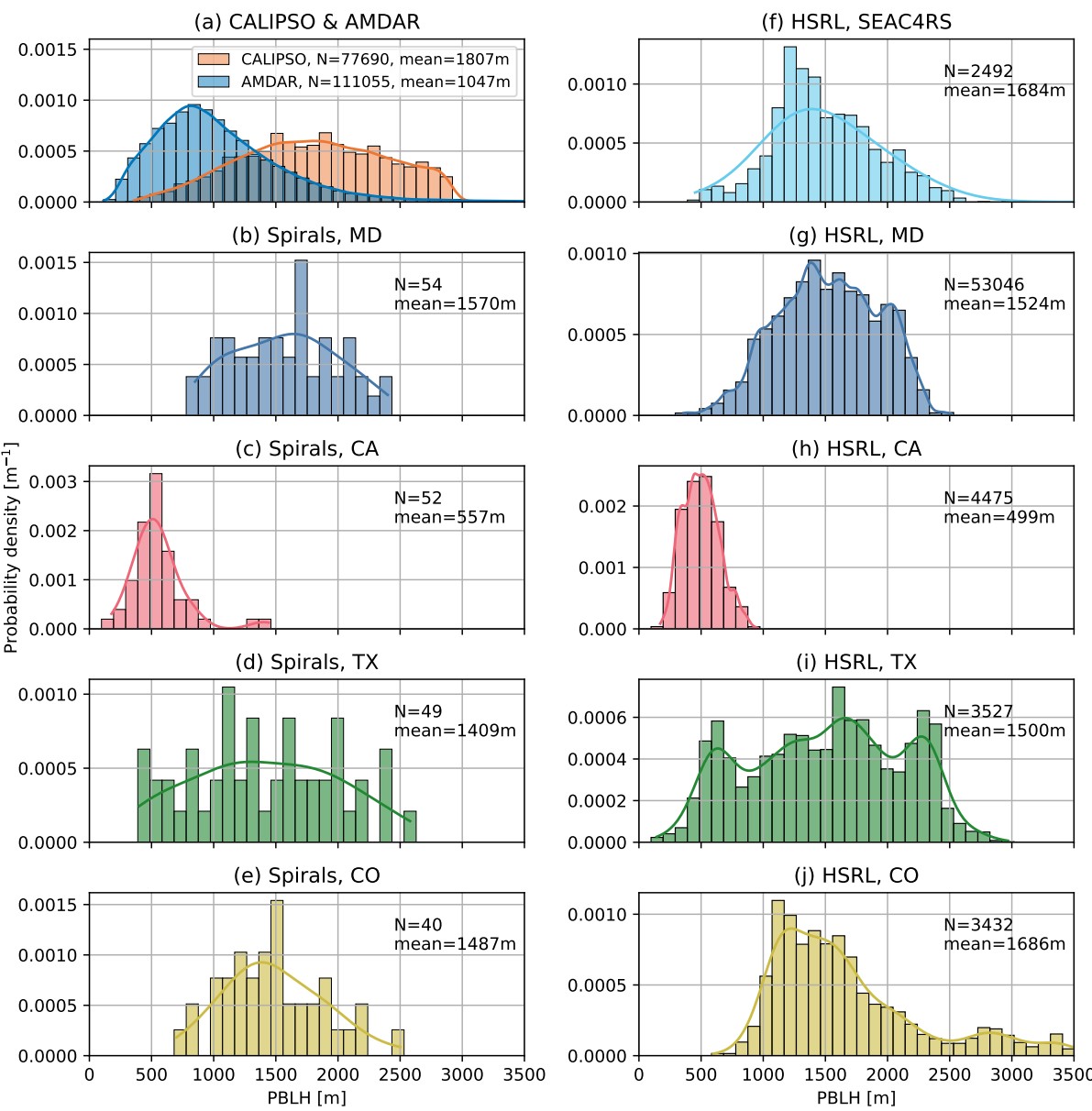

**Figure 3.** (a) Distributions of AMDAR PBLH after quantile regression-based filtering and CALIPSO PBLH over the CONUS. (b-e) Distributions of PBLH manually labeled using spiral profiles during the DISCOVER-AQ campaigns. (f-j) Distributions of PBLH from HSRL during the SEAC4RS and DISCOVER-AQ campaigns. The airborne campaign data were limited to 1230–1430 LST. The bins denote the normalized histograms and the lines denote smoothed kernel density estimations. The total numbers of PBLH data points and mean values are included in each panel.

where $\mathbf{y}$ is the response variable, or AMDAR PBLH here, $\mathbf{f}$ is an unknown function that generates response values when predictors are given, $\mathbf{X}$ is the predictor matrix, each column of which is a predictor, and $\mathbf{e}$ is the error term representing the model-observation mismatch. The learning algorithm tries to estimate the unknown function $\mathbf{f}$ by minimizing the sum of squared errors, $\mathbf{e}^T\mathbf{e}$, plus a regularization term to reduce overfitting. Various metrics related to the model performance can be calculated using $\mathbf{e}$ and $\mathbf{y}$, including the root mean squared error (RMSE), mean absolute error (MAE), and coefficient of determination ($R^2$). The trained model can then be used to make predictions at locations and times where AMDAR data do not exist but predictors are available.

## 3.1 Extreme gradient boosting (XGB) algorithm

In this work, the XGB algorithm (Chen and Guestrin, 2016) is used to estimate the predictive function $\mathbf{f}$. The XGB algorithm has been widely used in recent years to construct predictive models using complex environmental datasets (Keller et al., 2021; Masoudvaziri et al., 2021; Huang et al., 2022). We also found XGB features better performance and faster speed than random forest, another commonly used non-parametric machine learning algorithm. The linear regression model, although the simplest and fastest, was not used because of its slightly lower performance than XGB (testing RMSE higher by $\sim 5\%$) and random forest (testing RMSE higher by $\sim 3\%$) and the fact that the computing cost of XGB is not of concern. In a regression setting of the gradient boosting framework (Friedman, 2001), the model starts with a constant value as the prediction. A shallow regression tree is then added based on the residual of the previously predicted value and the tree's contribution is scaled by a learning rate to reduce the variance. Following this step, new residuals are calculated, another shallow regression tree is fitted to predict these new residuals, and the contribution of this tree is scaled as previous. This process is repeated until convergence. The XGB algorithm extends the normal gradient boosting by an optimized implementation.

## 3.2 Model hyperparameter tuning

The hyperparameters are external to the model and set before the learning process begins. They are tunable and can directly affect the model performance. We found that the most sensitive hyperparameters to the XGB model were the learning rate, the number of gradient boosted trees, and the maximum tree depth. Since the model training was relatively fast, we kept the learning rate to be a small value of $0.01$. Then the number of trees and the maximum tree depth were jointly optimized through a cross-validated grid search approach. The combinations of number of trees ranging from 200 to 1000 with a step of 100 and maximum tree depth ranging from 5 to 9 with a step of 1 were evaluated in the grid search. The cross validation was conducted by training the XGB model using data from 46 airports randomly selected from the total 54 airports in the AMDAR data, and the remaining 8 airports for each selection were used for testing. Such training vs. testing datasets split was repeated 100 times with the same random selector for each combination of hyperparameters. The average RMSE in the testing dataset was used as the metric. We found 800 as the number of trees and 8 as the maximum tree depth gave the best model performance. Although driven by the data, this selection yields a complicated XGB model. Since the main motivation of this study is to fill the spatial gaps between AMDAR sites over the CONUS during the AMDAR period without further extrapolating in space or time, we put more weight on the model performance than the simplicity or computational cost of model. The MAE, RMSE, and $R^2$

metrics using the optimized XGB hyperparameters on the training (46 airports randomly selected 100 times) and testing (the rest 8 airports) datasets are shown in Fig. 4. The metrics on the testing datasets show higher variance mostly because the testing datasets contain much less data points than the training datasets. The overall performances of the trained model on the unseen testing datasets are only slightly lower than on the training datasets. Specifically, the $R^2$ values indicate that on average, the model is capable of explaining $83\%$ variation in the training datasets and $75\%$ variation in the testing datasets. This lends confidence to the model capability to predict PBLH at locations with no AMDAR data coverage.

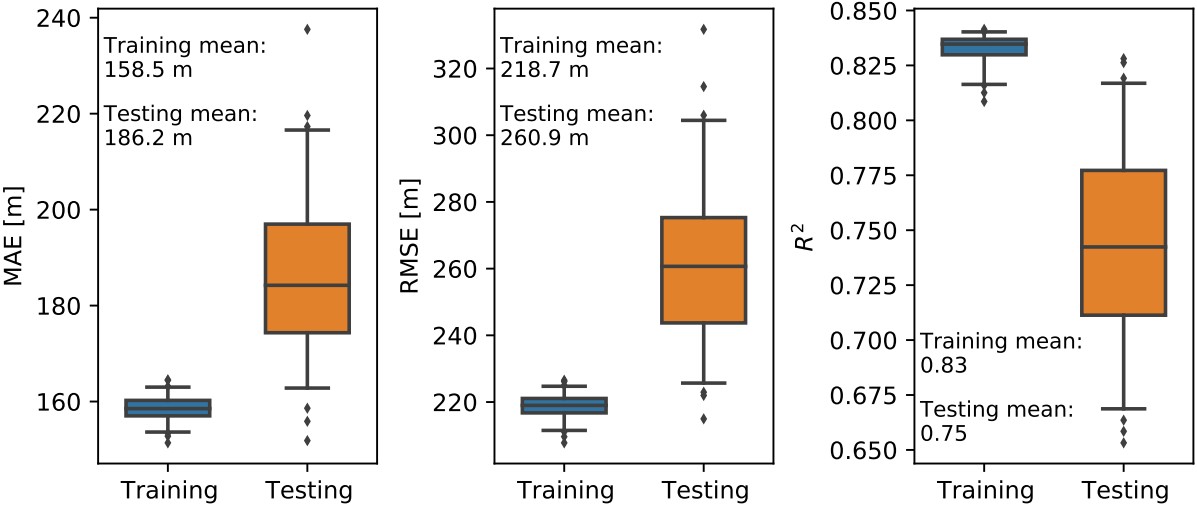

**Figure 4.** Box-whisker plots of the metrics, including MAE (left), RMSE (middle), and $R^2$ (right), of XGB model on the training and testing datasets. The horizontal lines show the 2.5, 25, 50, 75, and 97.5 percentiles, and the mean values are labeled in the plot.

### 3.3 Predictor selection

A range of meteorological data fields that might contribute to predicting the AMDAR PBLH were initially selected from the ERA5 reanalysis, including its own PBLH (named as boundary layer height in ERA5), surface pressure, trapping layer base height, 2-m temperature, skin temperature, cloud base height, evaporation, friction velocity, surface latent/sensible heat fluxes, surface net solar radiation, total precipitation, vertical integral of total energy, vertically integrated moisture divergence, and 100-m wind speed. Because the impacts of these environmental factors on the PBLH may not be instantaneous, these data fields were also sampled at 1, 2, and 3 hours prior to the AMDAR observation time. Although the same meteorological data fields can be sampled from other model or reanalysis datasets, we found that ERA5 in general gives the best performance. Besides the meteorological factors, geographical factors were also interpolated spatially and added as predictors, including the surface elevation (Danielson and Gesch, 2011), distance to the nearest ocean coastline (NASA, 2022), and estimation of land fraction in the vicinity from the $0.25 \times 0.3125°$ land fraction field from the GEOS-FP model (Lucchesi, 2018). In addition, we included the year and the day of year in the predictors to account for systematic observation-model mismatches with seasonal

and inter-annual variations. In total these add up to nearly 100 predictors. Although increasing the number of predictors will usually enhance the prediction power of the model, it is also desirable to use a parsimonious model that keeps predictors that make the most physical sense. To this end, we compared the relative importance of candidate predictors by running an inclusive model with all predictors using the permutation importance and the SHapely Additive exPlanations (SHAP) approaches. The permutation feature importance is defined to be the decrease in the model RMSE when the tabulated values from a single predictor are randomly shuffled (Pedregosa et al., 2011). This procedure breaks the relationship between the predictor and the response and thus quantifies how the model depends on the predictor. One drawback is that the permutation importance from a predictor may be underestimated if it is strongly correlated with other predictors. Correlation between predictors is significant especially between the same meteorological data at different lag hours. As such, we also incorporated the results from the SHAP approach that computes the contribution of each feature to the prediction based on coalitional game theory while considering feature interactions (Lundberg and Lee, 2017). Figure 5 shows the relative importance of the top 20 predictors calculated from these two approaches. Although the two algorithms work with different principles, the predictors identified with leading contributions to the overall model predicting power are generally consistent. We selected 14 predictors highlighted by red color in Fig. 5 in the final model. These include the boundary layer height, 2-m temperature, sensible heat flux, skin temperature, surface net solar radiation, distance to the ocean, land fraction, elevation, and year. The significance of year as a predictor indicates interannual variations that cannot be explained by other physics-based predictors. Therefore, this model should be used during the years when AMDAR data are available. Besides the concurrent hour boundary layer height, the boundary layer height data up to 3 hours prior were included, and the 2-m temperature and surface sensible heat flux 1 hour prior were also included.

## 3.4  Prediction of PBLH over the CONUS

The final model with selected hyperparameters and predictors were trained on the entire AMDAR dataset. Then the trained model was used to predict daily PBLH at 1300–1400 LST throughout the 2005–2019 period. The prediction was made on the ERA5 grid points as most predictors were already available on the same grid. Figure 6 compares the predicted daily early afternoon PBLH in March, April, May (MAM), June, July, August (JJA), September, October, November (SON), and December, January, February (DJF) in the top row with the corresponding values from ERA5, MERRA-2, and NARR in the following rows. The predictions from the XGB model trained on the AMDAR data show very similar seasonal spatial and seasonal variation patterns to the PBLH sampled from reanalysis (ERA5, MERRA-2, and NARR). The PBLH is significantly larger in the warm season than the cold season, and the intermoutain west and the southwest feature larger PBLH than other regions. The XGB model predicted PBLH is lower than the PBLH from three reanalysis in all four seasons. This result is consistent with the direct comparison between AMDAR PBLH and collocated ERA5 PBLH made by Zhang et al. (2020). We also look into finer-grained intercomparisons by averaging over nine different climate regions in the CONUS (Karl and Koss, 1984) at weekly resolution (i.e., weekly averages of PBLH at 1300–1400 LST). The results are shown in Fig. A1 and consistent with the seasonal maps shown in Fig. 6.

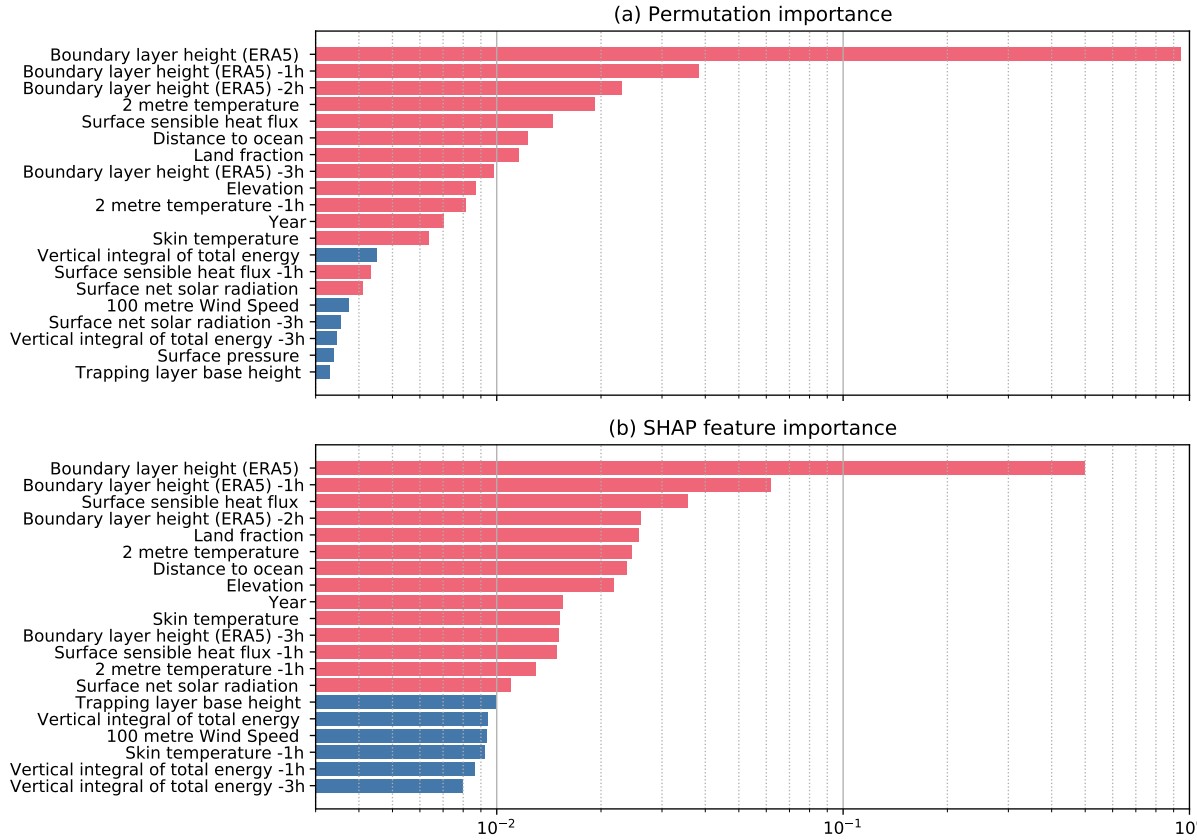

**Figure 5.** The feature importance values calculated by permutation importance (a) and the SHAP approach (b). All initial predictors were included in the calculation, and only the top 20 predictors are shown. Red-colored bars denote the predictors included in the final model.

## 4  Evaluation of predicted PBLH

Because the locations of AMDAR airports are sparse over the CONUS (Fig. 1a), it is important to evaluate the performance of the XGB model trained by AMDAR data. In addition to the XGB predicted PBLH, we also used spatiotemporally interpolated PBLH fields from NARR, MERRA-2, and ERA5 and included them in the evaluations against independent observational datasets.

### 4.1  Predicted PBLH vs. CALIPSO

The agreement between CALIPSO PBLH and the four datasets to be evaluated (NARR, MERRA-2, and ERA5 reanalyses as well as the XGB prediction) is strongly dependent on the ranges of PBLH. Figure 7 compares the mean bias (MB), MAE, RMSE, and Pearson correlation coefficient ($r$) between CALIPSO and these datasets with the comparisons split for CALIPSO PBLH at 0–1 km, 1–2 km, and 2–3 km. The dataset with the best performance (MB closest to 0, lowest MAE and RMSE, and

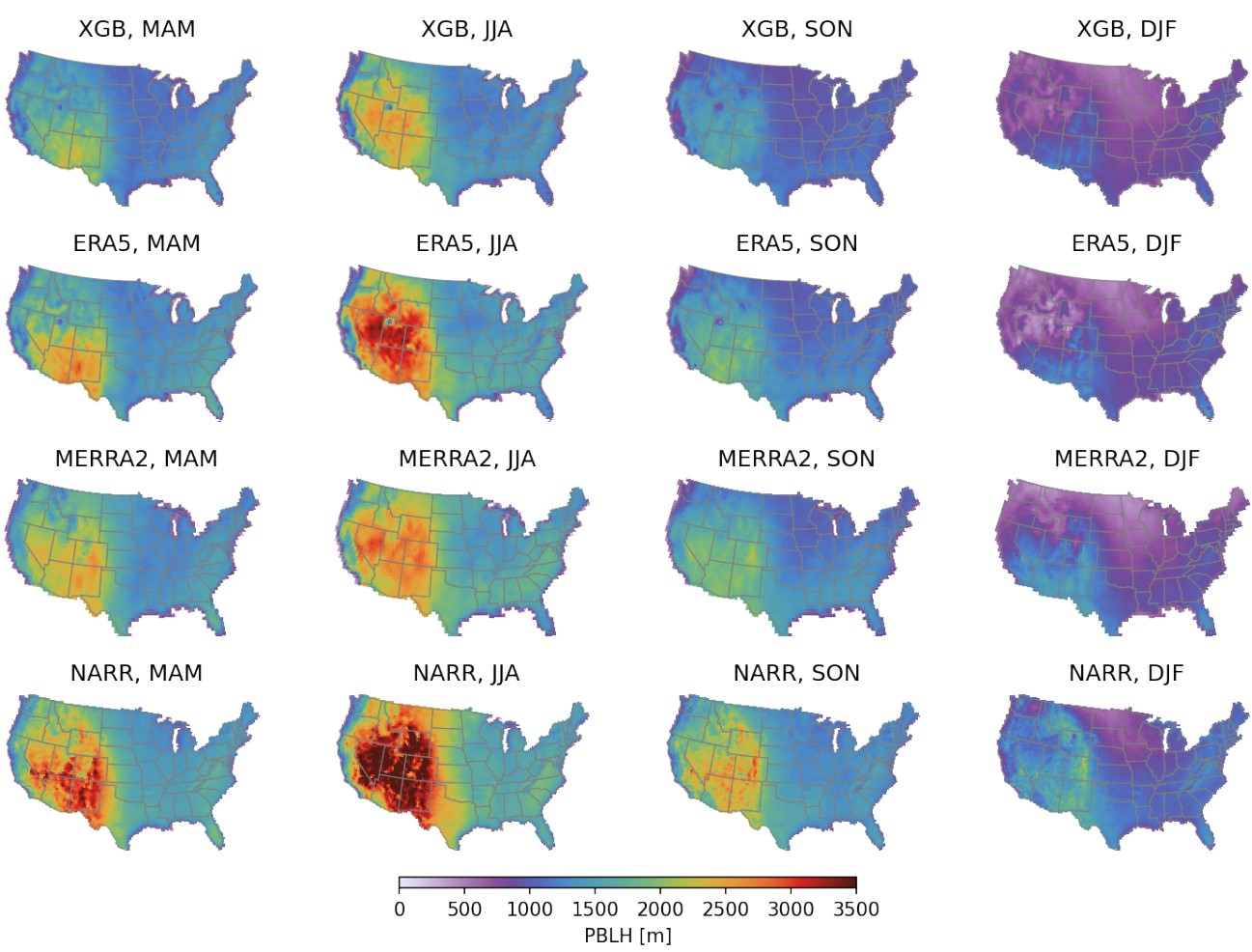

**Figure 6.** PBLH values at 1300–1400 LST predicted by the XGB model and the three corresponding PBLH values from ERA5, MERRA-2 and NARR. Daily values in 2005–2019 were averaged into four seasons as denoted in the titles.

highest $r$) is marked with a red star. When the CALIPSO PBLH is below 1 km, there is little correlation between CALIPSO and any other datasets (Fig. 7d), indicating that the CALIPSO retrieval might be unreliable at low PBLH. Furthermore, the overall

agreement for the 2–3 km range is lower than the counterpart for the 1–2 km range. The distribution of CALIPSO PBLH, as shown in Fig. 3a, is quite different from that of AMDAR PBLH, with a larger contribution in the 2-3 km range. This may explain the worst performance by the XGB prediction when CALIPSO PBLH is at 2-3 km by the lack of training AMDAR data in this range. However, the performance of the three reanalysis datasets is also counter-intuitive, as the NARR PBLH is one of the top performers here but is the least accurate in the following comparisons with airborne data (see below) . This lack

of coherence among datasets for PBLH above 2 km suggests CALIPSO retrieval might be also subject to systematic errors at these high values. As such, we focus on CALIPSO PBLH in the range of 1–2 km. In this range, the XGB prediction shows the best RMSE and correlation, competitive MAE, and a negative bias relative to CALIPSO. Moreover, the XGB prediction outperforms ERA5 in terms of MAE, RMSE, and correlation coefficient, demonstrating the effectiveness of model training using AMDAR data.

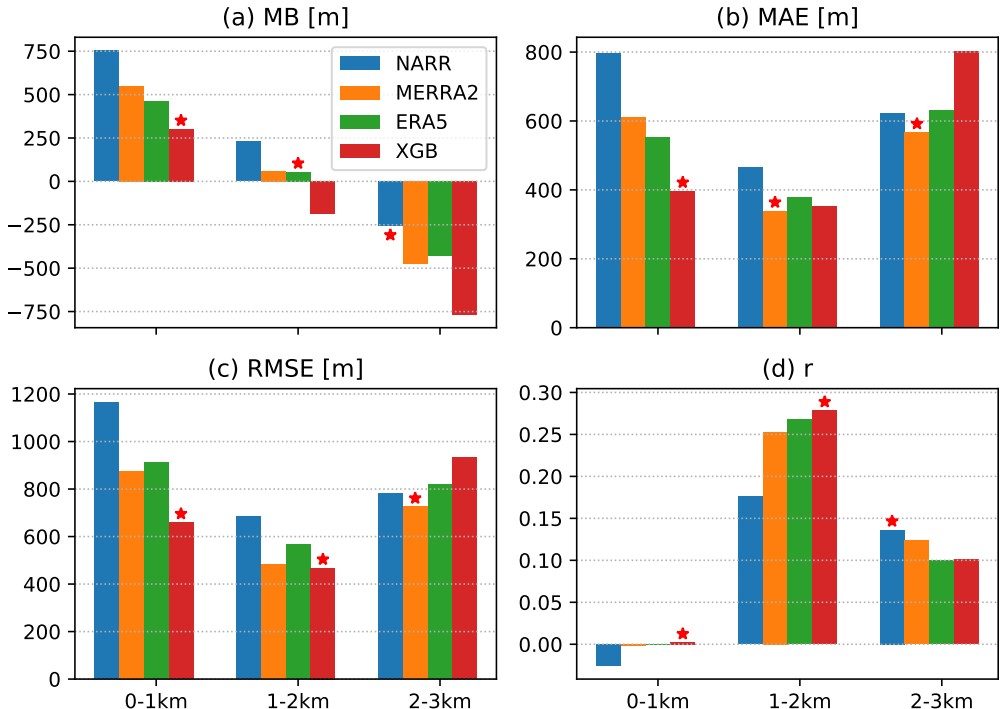

**Figure 7.** Agreements between CALIPSO retrieved PBLH and spatiotemporally collocated PBLH from NARR, MERRA-2, ERA5, and the XGB prediction. The CALIPSO dataset was split into ranges 0–1 km, 1–2 km, and 2–3 km, corresponding to the three clusters of bars in each panel. The evaluation metrics include MB (a), MAE (b), RMSE (c), and Pearson correlation coefficient (d). The red star labels dataset with the closest agreement.

A key strength of the CALIPSO dataset is its relatively uniform spatial distribution (Fig. 1b), which may help identify spatial patterns in the model prediction bias, especially in regions where AMDAR coverage is low. Figure 8 maps the bias between the four evaluated datasets and CALIPSO, binned in $2 \times 2.5°$ grid boxes over the CONUS. The comparisons were limited to cases with CALIPSO PBLH of 1–2 km, corresponding to the middle cluster of bars in Fig. 7. All four evaluated datasets show a high bias relative to CALIPSO in the west CONUS, except near the coastline. This high bias is the largest in

NARR and minimal in the XGB prediction. These high biases in the reanalysis PBLH were also noted by previous comparison studies (McGrath-Spangler and Denning, 2012; Zhang et al., 2020). The reanalysis datasets are in general agreement with CALIPSO in low-elevation regions in the east, whereas the XGB predictions show a near-uniform low bias of 100–200 m. This contributes to an overall low bias of XGB prediction relative to CALIPSO (Fig. 7a, middle cluster). As shown in the following comparisons with airborne data, the MB values of the XGB prediction relative to the reference datasets are generally

lower than the reanalysis datasets and are closer to zero. Hence a possible explanation is that CALIPSO is slightly biased high due to aerosol distributions above the PBL top. The spatial distribution of the XGB prediction bias relative to CALIPSO does not show more significant systematic features than the biases of the three reanalysis datasets, indicating that the spatial extrapolation of AMDAR PBLH through the meteorological and geographical features does not introduce excessive errors.

## 4.2   Predicted PBLH vs. HSRL

The same metrics were used to evaluate against the HSRL measurements during the DISCOVER-AQ and SEAC4RS campaigns, as shown in Fig. 9. The XGB prediction shows the lower MB than all the reanalysis datasets in four out of five campaigns with DISCOVER-AQ TX the only exception. The reanalysis datasets show large positive biases relative to HSRL in DISCOVER-AQ CO and SEAC4CRS, likely due to the large-scale positive biases in the west CONUS seen in Fig. 8a-c. The XGB prediction gives the best performance in four out of five campaigns (MD, CA, CO, and SEAC4CRS) in terms of MAE,

three out of five campaigns (MD, CO, and SEAC4CRS) in terms of RMSE, and three out of five campaigns (MD, TX, and CO) in terms of correlation coefficient. The ERA5 is the top performer for all cases where the XGB prediction is not at the top. Overall, ERA5 PBLH demonstrates remarkably better agreement with observations than NARR and MERRA-2. The performance of XGB prediction is robustly improved over ERA5 during DISCOVER-AQ MD, attributable to the high airport density in the northeast (Fig. 1a). The XGB prediction, however, does not significantly improve over ERA5 during DISCOVER-AQ

CA. A likely reason is that the airborne sampling was limited to the San Joaquin valley where no AMDAR data were available from Zhang et al. (2019).

        Compared with the evaluation using CALIPSO PBLH in the previous section, the correlations between the evaluated datasets and airborne HSRL LIDAR retrieved PBLH are substantially better. The correlation coefficient between XGB prediction and HSRL during DISCOVER-AQ TX is 0.8, while the HSRL PBLH spans a broad range from below 1 km to 3 km (Fig. 3i). The

correlation coefficients using CALIPSO are below 0.3 for all cases (Fig. 7d). Therefore, although both HSRL and CALIPSO PBLH are retrieved from aerosol backscatter gradients, the HSRL dataset is likely to have significantly better quality.

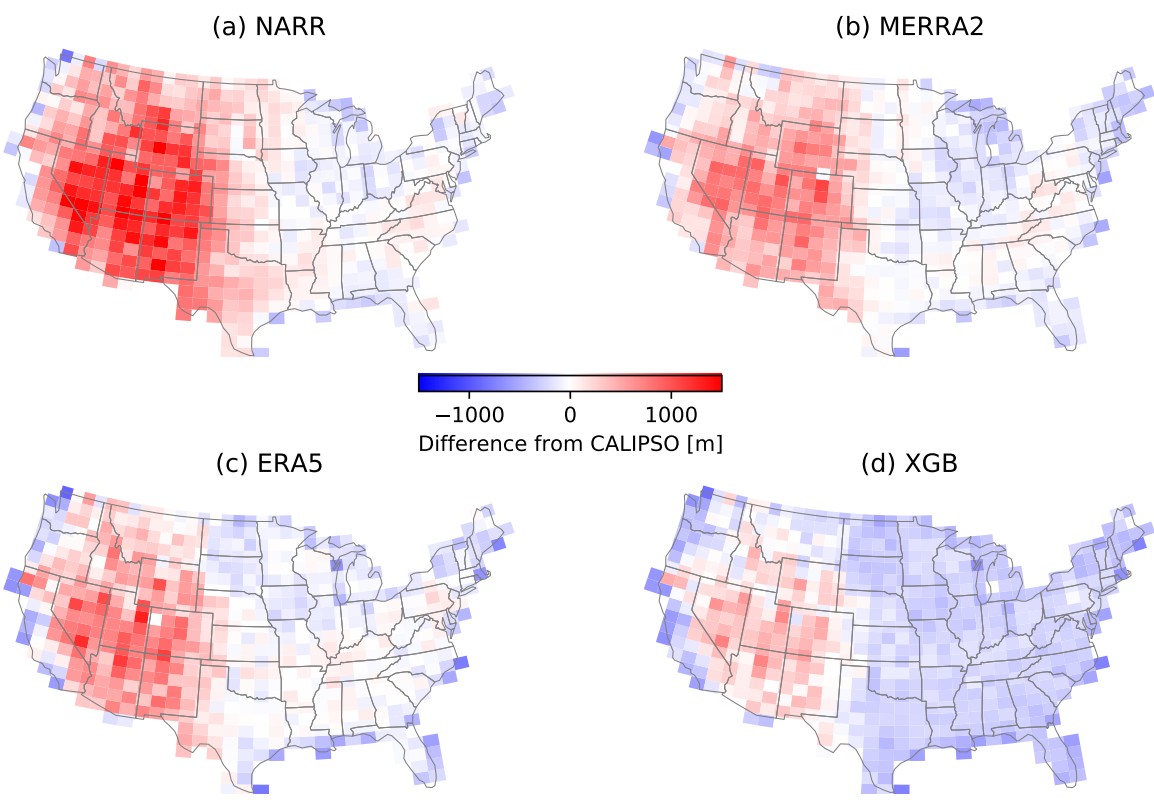

**Figure 8.** Gridded maps of the mean biases of NARR (a), MERRA-2 (b), ERA5 (c), and XGB prediction (d) relative to CALIPSO for all available soundings of CALIPSO in 2006–2013 with PBLH in 1–2 km. The spatial resolution is $2 \times 2.5°$.

## 4.3 Predicted PBLH vs. spiral profiles

During the four DISCOVER-AQ campaigns, PBLH was manually labeled from spiral profiles measured in-situ by aircraft. The comparisons between this PBLH observation dataset and the four evaluated datasets (NARR, MERRA-2, ERA5, and XGB prediction) are very similar to the comparisons using the HSRL dataset and shown in Fig. 10. This similarity is consistent with the resemblance of the distributions of the spiral profile- and HSRL-based PBLH (comparing the left and right columns in Fig. 3). The XGB prediction gives the best MB in all campaigns, the lowest MAE and RMSE in all campaigns but DISCOVER-AQ TX, and the highest correlation coefficient in all campaigns but DISCOVER-AQ CA.

Overall, the XGB prediction shows significantly better agreement with the three independent observational datasets than the reanalysis datasets, within which the ERA5 performs better than MERRA-2 and NARR. The XGB prediction outperforms ERA5 in most cases, mostly notably in DISCOVER-AQ MD where AMDAR airports are relatively dense.

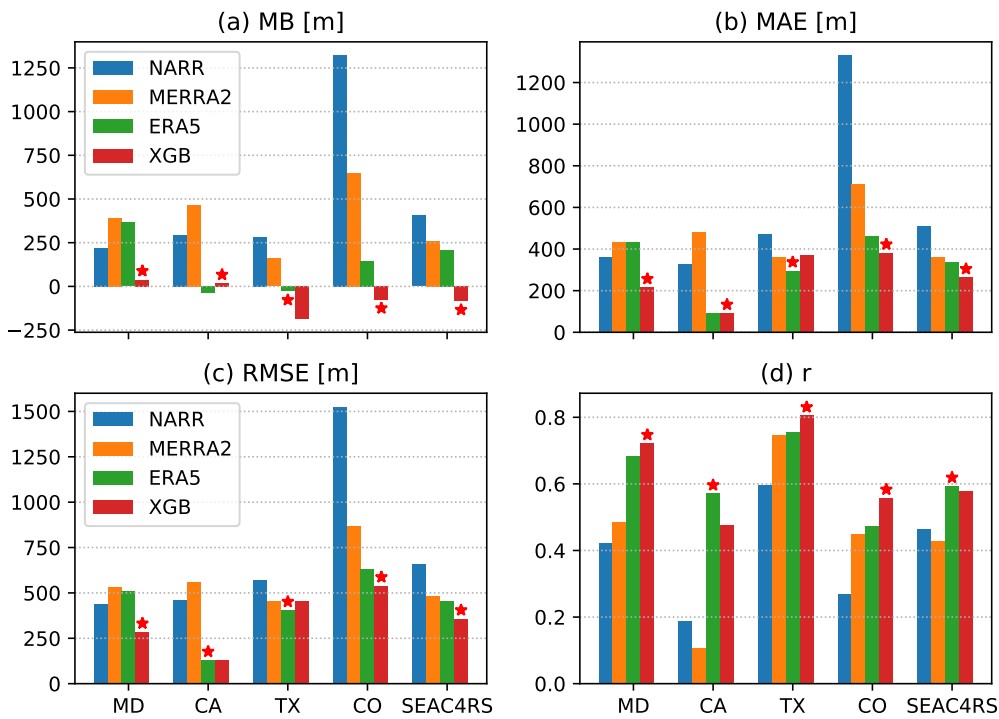

**Figure 9.** Similar to Fig. 7 but using HSRL PBLH measurements during five airborne campaigns as the reference dataset. Each campaign corresponds to a cluster of bars in each panel.

## 5 Conclusions

We developed a data-driven machine learning model to predict spatially complete PBLH over the CONUS daily at 1300–1400 LST using AMDAR data at 54 airport locations from 2005 to 2019 (Zhang et al., 2019). The XGB algorithm was used to train the regression model with predictors mostly selected from the ERA5 reanalysis. The model hyperparameters were optimized through cross-validated grid search by randomly splitting the airports into training and testing datasets. The predicted PBLH was evaluated using independent PBLH observations from CALIPSO, HSRL, and spiral profiles during the DISCOVER-AQ campaigns, with the caveat that different PBLH products and estimates have different ways of computing the PBLH and that PBLH observations used in these evaluation are sparse and subject to various uncertainties and inconsistency in retrieval methodology. The predicted PBLH is generally in better agreement with the evaluation datasets than PBLH sampled from reanalysis (NARR, MERRA-2, and ERA5). The reanalysis datasets give higher PBLH in the west CONUS relative to the evaluation datasets.

While this work demonstrates the potential use of data-driven approaches in deriving spatially complete PBLH, significant challenges still exist due to the uncertainty of existing datasets. We observe clusters of AMDAR observations that are uncorrelated with collocated ERA5 PBLH, mostly under stable conditions, and we find that no meteorological or geographical factors

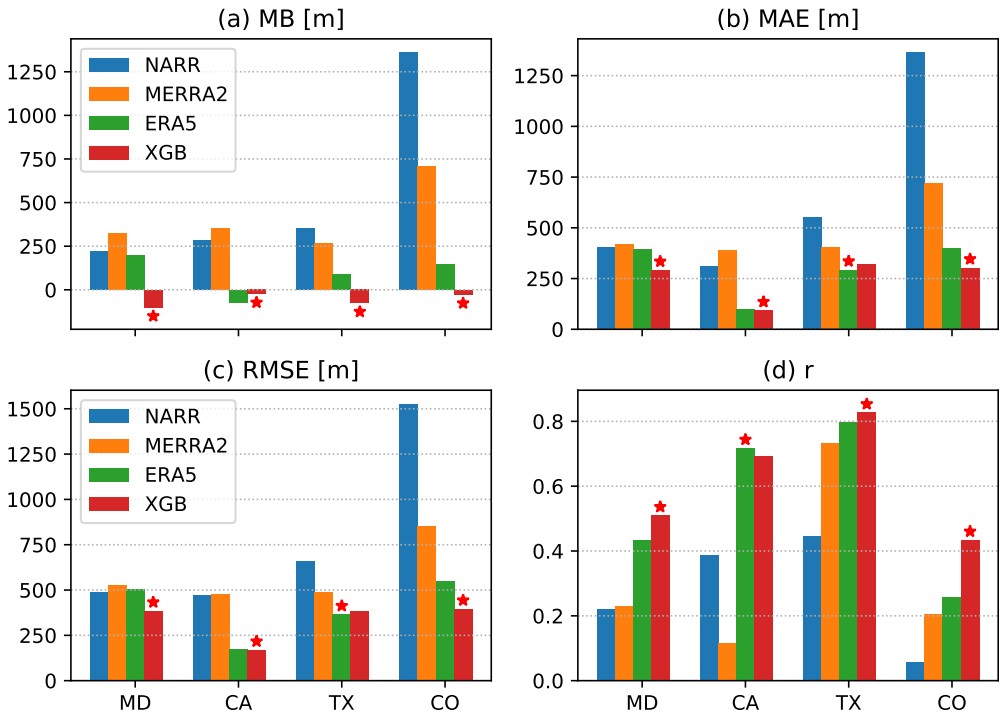

**Figure 10.** Similar to Fig. 9 but using PBLH labeled from spiral profiles during four DISCOVER-AQ campaigns.

could explain this discrepancy. A preprocessing step had to be implemented to mitigate its impacts on the model training. Moreover, the model configurations have been specifically optimized and evaluated to generate the spatially complete PBLH dataset over the CONUS during the period when AMDAR PBLH data are available (2005–2019). Further generalization of the work will require additional tuning and model evaluation. Also note that here our modeling focused on the entire CONUS.

Further improvements of model performance may be achieved by focusing on smaller geographic regions and fine tuning region-specific predictors.

The satellite-based CALIPSO dataset is the most spatiotemporally complete for model evaluation, but it is subject to large uncertainties, which gives essentially no correlations with reanalysis datasets and the prediction from this work when PBLH is lower than 1 km. Future spaceborne PBLH observations with higher fidelity and more routine suborbital measurements,

especially under stable conditions, will be beneficial.

This work extends the AMDAR-based PBLH data from Zhang et al. (2020) that are only available at point locations to the entire CONUS, enabling sampling PBLH at the sounding locations and times of low-earth orbit spaceborne sensors with over pass time in the early afternoon. Since the AMDAR PBLH data are available hourly, it is possible to extend this work to other daytime hours and even nighttime hours with the caution that it will be more challenging due to the increase of stable conditions

and less observational datasets available for evaluation. The HSRL and spiral profile PBLH observations will also be available

in other daytime hours but not CALIPSO. This extension will synergize with satellite atmospheric composition observations in the morning orbits and the TEMPO mission that covers the North America at hourly resolution (Zoogman et al., 2017).

*Data availability.* The AMDAR data are available through the Meteorological Assimilation Data Ingest System web service portal at https://madis-data.cprk.ncep.noaa.gov/madisPublic1/data/archive/. The hourly boundary layer profiles and PBLH based on AMDAR data can be found at 10.5281/zenodo.3934378. The ERA5 data are available from the Copernicus Climate Data Store at https://cds.climate.copernicus.eu/. The MERRA-2 data are available through the NASA GES DISC at https://disc.gsfc.nasa.gov/datasets/M2T1NXSLV_5.12.4/summary. The NARR data are available from the NCAR Research Data Archive at https://rda.ucar.edu/datasets/ds608.0/. The CALIPSO data are available through the University of Wisconsin-Madison Space Science and Engineering Center website (https://download.ssec.wisc.edu/files/calipso/). The DISCOVER-AQ data are available at http://doi.org/10.5067/Aircraft/DISCOVER-AQ/Aerosol-TraceGas and the SEAC4RS data are available at http://doi.org/10.5067/Aircraft/SEAC4RS/Aerosol-TraceGas-Cloud.

## Appendix A:  Weekly resolved PBLH comparison over different climate regions

*Author contributions.* ST, DL, and KS developed the methodology. ZA, ST, and KS performed the analysis and wrote the paper. ZA and KS managed the datasets. AJS provided expertise on airborne Lidar measurements. RK provided expertise on spaceborne Lidar measurements. All co-authors contributed to the editing of the paper.

*Competing interests.* The authors declare that they have no conflict of interest.

*Acknowledgements.* This research has been supported by the NASA Earth Science Division Atmospheric Composition: Modeling and Analysis Program (ACMAP, Award 80NSSC19K0988). We thank Mingchen Gao at University at Buffalo for helpful discussions.

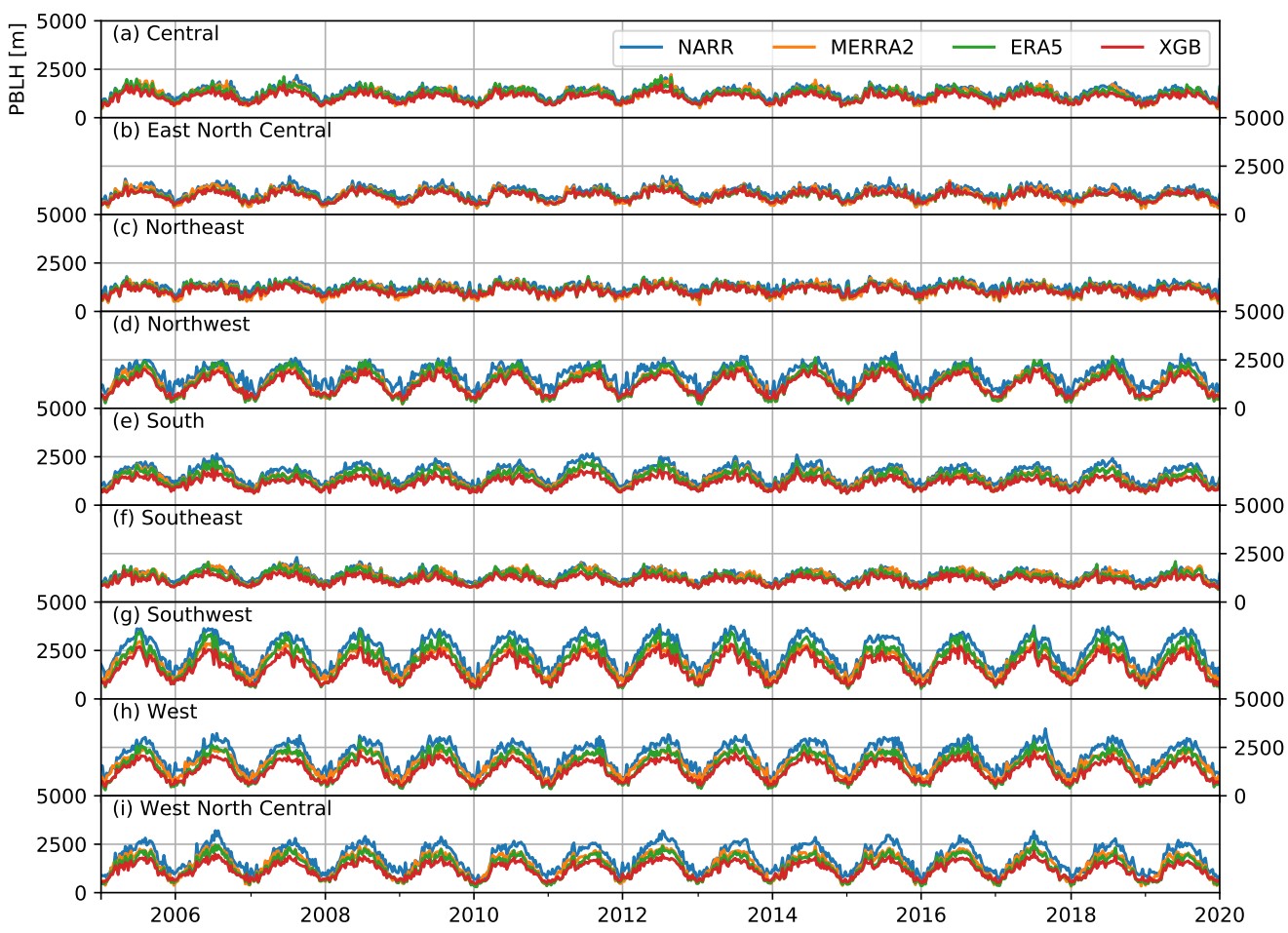

**Figure A1.** Weekly average PBLH at 1300–1400 LST over nine climate regions over the CONUS.

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
