# Peer review of "Estimates of spatially complete, observational data-driven planetary boundary layer height over the contiguous United States"

_Atmospheric Measurement Techniques, 2022_

## Author Comment (AC1)

We would like to thank the referee for the useful comments and constructive suggestions. In the following, we address the referee's comments and describe corresponding changes we have made to the manuscript. The referee's comments are listed in *italics*, followed by our response in blue. New/modified text in the manuscript is in **bold**.

*The main issue and limitation of this work is in the validation of this new PBLH product. Because it is novel, and there is such a need, there are not any observationally-based 'truth' products to wok with that don't each have their own large uncertainties and limitaitons. CALIPSO is used here as 'truth', but has significant issues in terms of estimating an automated PBLH products (dependent on PBL regime, signal to noise, elevated aerosol gradients, clouds, etc.). So the anlayses and intercomparisons are more relative than absolute, in terms of comparing the new AMDAR/ML PBLH vs. models (reanalyses) vs. observations (CALIPSO and aircraft-based lidar). As a sanity check, this comparison is useful, but it does not sufficiently reflect whether the new PBLH product is accurate on day-day or diurnal timescales.*

We agree with the referee that the PBLH products and estimates used in this work are different in many aspects, and thus their intercomparisons should be interpreted with caution. In this paper, we use CALIPSO and two other independent PBLH estimates (e.g., from research aircraft profiles and HSRL airborne lidar) as the "reference", but we do not necessarily view them, especially CALIPSO, as the "truth". Perhaps this point was not very clearly articulated in the previous submission. In the revised manuscript, we explicitly add a sentence in the abstract about this:

**"Compared with PBLHs from reanalysis products, the PBLH prediction from this work shows closer agreement with the reference observations, with the caveat that different PBLH products and estimates have different ways of identifying the PBLH and thus their comparisons should be interpreted with caution."**

In the introduction, we also revise the sentence at lines 73–75 to the following:

**"The predicted PBLH is then compared to PBLHs from three widely-used reanalysis products (ERA5, MERRA-2, and NARR), and all of them are further compared to independently diagnosed PBLH from observations (e.g., from research aircraft profiles and HSRL airborne lidar) and the CALIPSO PBLH product. Here we should emphasize that such comparisons should be interpreted with caution since different PBLH products and estimates have different ways of identifying the PBLH. In this sense, none of the PBLH products and estimates should be treated as the golden truth. However, such comparisons remain meaningful because (1) they provide confidence in the PBLH prediction by this work and (2) they generate information regarding the difference between various PBLH products and estimates that have been widely used in previous work.**

Furthermore, we add a new section (2.3.4) to discuss the pros and cons of the different

datasets (see the response to the following comment).

Lastly, the day-day accuracy of the PBLH product is evaluated by adding a new appendix figure, in response to the comment regarding Fig. 6 below. But since we focus on 13-14 local time, the diurnal timescales cannot be resolved by this study.

*Another concern is related to the diversity in the PBLH estimation techniques used in all these products that are being evaluated. L45 (paragraph). Can the authors say something about the methods used in these comparisons? Using CALIPSO implies that PBLH is based on aersol backscatter gradients, which is quite distinct from what each model PBLH is based upon thermodynamically (not to mention their respective PBLH approach that is mentioned earlier). Neither is actually 'wrong' or 'right', as they are looking for the top of the mixed layer or the top of the PBL turbulence, or T and RH gradients. The conclusions presented here suggest that CALIPSO aerosol-based PBLH is the 'truth' but also AMDAR-based thermodynamic PBLH is the 'truth', and both cannot be the case. These are relative intercomparisons that show that the models deviate from other estimates, but has it been shown when looking at T and q profiles that the models actually do 'overestimate' the true PBLH?*

Again, we fully agree with the referee that different PBLH products and estimates use different methods to identify the PBLH. But we would like to interpret this as a justification to our work. It is exactly because of the differences in the methodology that a comparison between them is meaningful (at least for some users who are only interested in the PBLH values, not how they are generated), with the caveat that none of them should be blindly interpreted as the truth, as we elaborated in our response to the previous comment. To address the reviewer's concern, in the revised manuscript we focus on (1) emphasizing that none of the PBLH products and estimates should be interpreted as the golden truth (see our response to the previous comment) and (2) articulating the differences between different datasets. In particular, we group the descriptions of the three evaluation datasets (CALIPSO, HSRL, and spirals) into a single subsection ("**2.3 Observational datasets used for evaluation**") and add a new subsubsection to overview the pros and cons of these datasets:

**"2.3.4 Comparisons of observational datasets**

**As summarized by Figs. 1 and 3, none of the observational datasets described above can uniformly represent the PBLH over the study domain. CALIPSO features the most homogeneous spatial coverage (Fig. 1b), but its PBLH product relies on an automatic, global algorithm that may be subject to significant uncertainties. Yet the unique benefit of including CALIPSO data is that it can indicate errors in the spatial prediction made by our model, as the availability of AMDAR airports is spatially clustered (Fig. 1a). For example, no AMDAR sites are available in the large area over the Northern Rockies and Plains and the Southeast. Because of the large differences in AMDAR and CALIPSO PBLH, we consider the intercomparison involving CALIPSO more relative than absolute and focus on correlations rather than biases.**

**One should also note that CALIPSO and HSRL PBLH data are based on aerosol**

**backscatter gradients, which is quite distinct from AMDAR, DISCOVER-AQ spiral profiles, and ERA5, where PBLH values are diagnosed thermodynamically. Although systematic differences between aerosol-based and thermodynamics-based PBLH may exist, we do not observe them by comparing spatiotemporally close spiral and HSRL measurements in the same DISCOVER-AQ campaigns (i.e., comparing d vs. g, c vs. h, d vs. i, and e vs. j in Fig. 3). Furthermore, the model prediction from this work may serve as a "traveling standard" when evaluated against HSRL and spiral datasets. As will be shown in Sections 4.2 and 4.3, the biases between HSRL data and collocated model prediction do not show significant differences from the biases between spiral data and the corresponding model prediction."**

We take the PBLH from the reanalysis datasets as is without any attempts to re-diagnose the PBLH values from raw profiles. We acknowledge the differences between models by revising the sentences in lines 336–337 to the following:

**"The predicted PBLH is generally in better agreement with the observational datasets than PBLH sampled from reanalysis (NARR, MERRA-2, and ERA5) with the caveat that PBLH observations used in these evaluation are sparse and subject to various uncertainties and inconsistency in retrieval methodology. The reanalysis datasets give higher PBLH in the west CONUS relative to observational datasets."**

*To reiterate the comment about the diversity of PBLH methods in the models and observations, what are the implicatons of comparing PBLH derived in 6 different ways (AMDAR, ERA, NARR, M2, plus CALIPSO, and HSRL)? There are tendencies from each method (TKE, RiB, etc.) in terms of the PBLH they capture and under what regimes they perform well/poorly. None is perfect, but comparing across all of them is problematic in a blanket sense.*

We fully agree with the referee that different PBLH products and estimates use different methods to identify the PBLH, but we do not necessarily agree that comparing across them is problematic. In fact, we would like interpret this as a justification to our work. It is exactly because of the differences in the methodology that a comparison between them is meaningful (at least for some users who are only interested in the PBLH values, not how they are generated), with the caveat that none of them should be blindly interpreted as the truth, as we elaborated in our response to the previous comments.

To address the reviewer's concern, in the revised manuscript we focus on (1) emphasizing that none of the PBLH products and estimates should be interpreted as the golden truth and (2) articulating the differences between different datasets. Details can be found in our response to the previous comments.

*Fig. 6: It would be nice to see a more detailed, nuanced analysis zooming into regions, and sub-seasons (even day-day). This generally looks like a good result for XGB, but as we know*

*a lot of important variability and biases can be masked out on the seasonal timescale. Could the authors provide this even if in supplemental form?*

We zoom into the nine US climate regions defined by NOAA and add the weekly averaged PBLH in the reanalysis datasets and XGB. The daily plots show very similar temporal pattern but give too much granularity for such long time series. The following figure is added as an appendix:

[Figure]

Figure 1: (Figure A1 in the revised manuscript) Weekly average PBLH at 1300–1400 LST over nine climate regions over the CONUS.

The following is added to the end of section 3.4:

**"We also look into finer-grained intercomparisons by averaging over nine different climate regions in the CONUS (Karl and Koss, 1984) at weekly resolution (i.e., weekly averages of PBLH at 1300–1400 LST). The results are shown in Fig. A1 and consistent with the seasonal maps shown in Fig. 6."**

*Section 5: Given the diversity in models, observations, and PBLH estimation approaches discussed above, it would be helpful to include a more direct analysis of individual profiles (not just seasonal composites, or CONUS evaluations). Examining individual profiles would enable a direct comparison across all of these, and a visual aid to actually look at T, q, and aerosol profiles and their estimated PBLH. This would provide insight as to their behavior, and also provide to the reader a 2D vertical perspective of what this paper is all about and how much these can differ. The challenge is in selecting/sampling the locations and times, but that could be done in a single figure with multiple panels, regions, etc. after the authors perform a search of different locations and regimes that they feel are representative. No overall conclusions would be made based on these, but it likely would provide the insight and demonstrate the variability to the reader. Might also expose what CALIPSO is doing.*

While we totally agree that directly comparing individual profiles can produce fundamental insights into the behaviors of different products and observations (e.g., we can compare the AMDAR and ERA5 profiles of T), we deem that comparing individual profiles will make the paper deviate from its central theme, which is to develop a model for predicting the PBLH over the entire CONUS based on AMDAR data. As the reviewer pointed out, different products and observations use different features to identify the PBLH (e.g., AMDAR and ERA5 use profiles of temperature and winds with or without friction velocity information, CALIPSO uses aerosol content). If we were to compare individual profiles, we would have to answer the question of how differences in these profiles translate to differences in the diagnosed PBLHs, which is a grand challenge and is not the intent of this work. Hence our strategy is to focus entirely on the PBLH without worrying too much about the exact features in the profiles leading to the diagnosed PBLH. Admittedly, this dodges many important/interesting complexities involved in identifying the PBLH, which is necessary for this work. But exploring the nuances in these comparisons could be the topic of a follow-up study.

Also we note that this work is a follow-up study of Zhang et al. (2019, 2020) where some aspects of the individual profiles from AMDAR and DISCOVER-AQ have been analyzed. For example, hourly PBLHs from individual profiles of AMDAR and DISCOVER-AQ are compared (see Figure 4 in Zhang et al. 2020).

*Intro: Strong aerosol and AQ motivation here in terms of PBLH. The authors could also mention importance of PBLH for convection, shallow to deep convection, LCL deficit, etc. in terms of the thermodynamic pathways and feedbacks (ultimately on precipitation). Also the PBL mediation of surface fluxes (H and LE), soil moisture, vegetation, entrainment feedbacks. There is a big component of PBLH importance that isn't discussed here and would make the general impact of this work and dataset more robust.*

We thank the referee for this very constructive suggestion. In the revised manuscript, the role of PBLH in land-atmosphere interaction is discussed in the start of the introduction:

**"The planetary boundary layer (PBL) is the lowest part of the atmosphere that mediates the exchange of momentum, energy, and mass between the surface**

**and the overlying free troposphere (Stull, 1988). It plays a central role in land-atmosphere coupling, linking surface states and characteristics (e.g., surface temperature, soil moisture, and vegetation) to convection through surface fluxes of sensible and latent heat (Santanello et al., 2018). Improving our understanding and characterization of the PBL is critical for enhancing the predictability of numerical weather prediction and global climate and earth system models (Garratt, 1994; Stensrud, 2009). The PBL height (PBLH), which characterizes the vertical extent of the PBL, is a critical parameter in many land-atmosphere coupling metrics (Santanello et al., 2013, 2015). The PBLH also governs the vertical mixing of thermal energy, water, and trace gases and hence strongly regulates the near-surface pollutant concentrations."**

*L36: Another reference that looked at PBLH in M2, NARR, and CFSR: https://doi.org/10.1175/JCLI-D-14-00680.1*

The reference is added in the previous response.

*L79: Also potential applicability to drifting orbits of existing satellites (e.g. EOS) that cover other parts of the diurnal cycle.*

This sentence is revised to:

**"Since both AMDAR and ERA5 are continuous at hourly resolution, future work may extend to other daytime hours observable by geostationary missions like TEMPO (Zoogman et al., 2017) and potentially to existing satellites with drifted orbits such as the NASA Earth Observing System (EOS) satellites."**

*L99: Many models and applicaitons have used 0.25. Can the authors justify using 0.5 here? A direct comparison of the results when using 0.5 vs. 0.25 seems important, given the sensitivty to the assumption and the impact on the ultimate product of PBLH.*

First of all, we note that both 0.25 and 0.5 have been used as empirical thresholds for the bulk Richardson number method for computing PBLH, as reviewed elsewhere (Zhang et al., 2014). While the 0.25 value stems from the Miles-Howard theory associated with stability of heterogeneous shear flow, the exact value of the critical gradient Richardson number, even whether a critical gradient Richardson number exists or not, remains debated (Galperin et al., 2007; Grachev et al., 2013; Li et al., 2015). In this sense, we deem that the 0.5 value, as an empirical value, is justified.

Second, the reason we choose 0.5 in this work follows the previous study by Zhang et al. (2020), who determined the critical Richardson number value (as well as two other parameters needed by this method) with an empirical calibration approach. For example, they compared AMDAR PBLH estimated with the critical Richardson number values of 0.5 and 0.25 and found that using 0.5 yields better estimates of PBLH compared to their reference datasets.

Finally, we note that ERA5 used 0.25 as another reviewer pointed out. However, the AMDAR

and ERA5 profiles are fundamentally different as one is measured in-situ, while the other is model-based and on a 0.25° grid. It is highly likely that AMDAR profiles contain structures that cannot be resolved by ERA5. Hence the optimal parameters for computing the PBLH are unlikely the same between AMDAR and ERA5, justifying the empirical approach taken by Zhang et al. (2020).

*Section 2.3: For CALIPSO, as McGrath-Spangler has shown, it is very difficult to automate PBLH retrieval due to elevated aerosols, clouds, signal to noise, etc. such that they were only able to confidenly produce seasonal climatologies. Because you are estimating your own CALIPSO-based PBLH here on a daily basis, there is likely much greater uncertainty due to these factors on the day-day level.*

Thanks for bringing this up. Although we used CALIPSO PBLH values on a per sounding basis, we evaluate the overall metrics like correlation and mean bias. We include these discussions in the newly added section 2.3.4.

*Fig. 5: I'm glad the authors included this detailed assessment of the variables more influential in the ML training. This is where the L-A interaction component becomes scientifically interesting and can be learned from. Fig. 5 makes sense (to me) overall in terms of which variables most impact/drive CBL growth, focused on surface heating and buoyancy (as well as the diurnal pattern/memory of the PBL growth itself). Identifying which is most land-driven (vs. water/ocean) makes sense as well. Given the focus of soil moisture in L-A interaction research, It would have been interesting to include soil moisture itself, given its role in controlling the Bowen ratio and surface heating (which are strongly infliential as seen here). Also, it is interesting that LHF doesn't emerge along with H. Using EF or Bowen ratio might have been a strong as well if included.*

We also note that the thermal factors tend to contribute more than the water-related ones, although this may not be a universal finding. The following is added after line 344:

**"Future improvements of model performance may be achieved by focusing on smaller geographic regions and fine tuning region-specific predictors."**

*XGB Training: Did the authors test the sensitivity of the training result to the dataset used (ERA5 vs. M2, for example)? As presented, the results are strongly dependent on the relationship of AMDAR vs. ERA5 in terms of PBLH.*

We tested GEOS-FP instead of MERRA-2, which features finer spatial resolution than MERRA2, comparable to ERA5. The performance is slightly lower than using predictors from ERA5. This is expected as the PBLH is the dominant predictor, and ERA5 PBLH is more consistent with the observational reference than MERRA-2 and NARR (Figs. 6-10). The following sentence is added to line 233:

**"Although the same meteorological data fields can be sampled from other model or reanalysis datasets, we found that ERA5 in general gives the best performance."**

*L319: 'Complement' CALIPSO(?), or is the HSRL superior in terms of identifying a robust PBLH? Does this put the quality of CALIPSO PBLH in question? There may be something to say here about both estimates being based on aerosol backscatter gradients, but yielding such different results vs. XGB.*

We agree that while both are based on aerosol backscatter gradients, HSRL shows significantly better quality than CALIPSO. Nonetheless, CALIPSO gives a much more uniform spatial and temporal coverage. This sentence is revised to the following:

**"Therefore, although both HSRL and CALIPSO PBLH are retrieved from aerosol backscatter gradients, the HSRL dataset shows significantly better quality."**

*Section 4.3: This is also interesting, in that the HSRL is based on aersol backscatter, but the spirals are manually based on other variables, yet yield similar results that are quite different from CALIPSO. Again, I would suggesting considering more rigorously the uncertainty in the CALIPSO estimates.*

We also note this point in the newly added section 2.3.4.

*Conclusions: Rather short conclusions that could offer more applicability to the community, and what this dataset could be used for (and how confidently), as well as what new PBL profile observations would be most valuable in the future (e.g. spaceborne or more routine suborbital measurements) to reduce the uncertainty in the estimates applied here.*

We add the following as the second paragraph of the conclusion:

**"Significant challenges still exist due to the lack of PBLH observations and the uncertainty of existing datasets. We observe clusters of AMDAR observations that are uncorrelated with collocated ERA5 PBLH, mostly under stable conditions, and no meteorological or geographical factors could explain this discrepancy. A preprocessing step had to be implemented to mitigate its impacts on the model training. The satellite-based CALIPSO dataset is the most spatiotemporally complete for model evaluation, but it is subject to large uncertainties, which gives essentially no correlations with reanalysis datasets and the prediction from this work when PBLH is lower than 1 km. Future spaceborne PBLH observations with higher fidelity and more routine suborbital measurements, especially under stable conditions, will be beneficial."**

**References**

Galperin, B., Sukoriansky, S., and Anderson, P. S.: On the critical Richardson number in stably stratified turbulence, Atmospheric Science Letters, 8, 65–69, 2007.

Garratt, J. R.: The atmospheric boundary layer, Earth-Science Reviews, 37, 89–134, 1994.

Grachev, A. A., Andreas, E. L., Fairall, C. W., Guest, P. S., and Persson, P. O. G.: The

critical Richardson number and limits of applicability of local similarity theory in the stable boundary layer, Boundary-layer meteorology, 147, 51–82, 2013.

Karl, T. and Koss, W. J.: Regional and national monthly, seasonal, and annual temperature weighted by area, 1895-1983, URL `https://repository.library.noaa.gov/view/noaa/10238`, 1984.

Li, D., Katul, G. G., and Bou-Zeid, E.: Turbulent energy spectra and cospectra of momentum and heat fluxes in the stable atmospheric surface layer, Boundary-layer meteorology, 157, 1–21, 2015.

Santanello, J. A., Peters-Lidard, C. D., Kennedy, A., and Kumar, S. V.: Diagnosing the nature of land–atmosphere coupling: A case study of dry/wet extremes in the US southern Great Plains, Journal of Hydrometeorology, 14, 3–24, 2013.

Santanello, J. A., Roundy, J., and Dirmeyer, P. A.: Quantifying the land–atmosphere coupling behavior in modern reanalysis products over the US Southern Great Plains, Journal of Climate, 28, 5813–5829, 2015.

Santanello, J. A., Dirmeyer, P. A., Ferguson, C. R., Findell, K. L., Tawfik, A. B., Berg, A., Ek, M., Gentine, P., Guillod, B. P., Van Heerwaarden, C., et al.: Land–atmosphere interactions: The LoCo perspective, Bulletin of the American Meteorological Society, 99, 1253–1272, 2018.

Stensrud, D. J.: Parameterization schemes: keys to understanding numerical weather prediction models, Cambridge University Press, 2009.

Stull, R. B.: An introduction to boundary layer meteorology, Kluwer Academic, 1988.

Zhang, Y., Gao, Z., Li, D., Li, Y., Zhang, N., Zhao, X., and Chen, J.: On the computation of planetary boundary-layer height using the bulk Richardson number method, Geoscientific Model Development, 7, 2599–2611, 2014.

Zhang, Y., Li, D., Lin, Z., Santanello Jr, J. A., and Gao, Z.: Development and evaluation of a long-term data record of planetary boundary layer profiles from aircraft meteorological reports, Journal of Geophysical Research: Atmospheres, 124, 2008–2030, 2019.

Zhang, Y., Sun, K., Gao, Z., Pan, Z., Shook, M. A., and Li, D.: Diurnal Climatology of Planetary Boundary Layer Height Over the Contiguous United States Derived From AMDAR and Reanalysis Data, Journal of Geophysical Research: Atmospheres, 125, e2020JD032 803, https://doi.org/https://doi.org/10.1029/2020JD032803, e2020JD032803 2020JD032803, 2020.

Zoogman, P., Liu, X., Suleiman, R., Pennington, W., Flittner, D., Al-Saadi, J., Hilton, B., Nicks, D., Newchurch, M., Carr, J., Janz, S., Andraschko, M., Arola, A., Baker, B., Canova, B., Chan Miller, C., Cohen, R., Davis, J., Dussault, M., Edwards, D., Fishman,

J., Ghulam, A., González Abad, G., Grutter, M., Herman, J., Houck, J., Jacob, D., Joiner, J., Kerridge, B., Kim, J., Krotkov, N., Lamsal, L., Li, C., Lindfors, A., Martin, R., McElroy, C., McLinden, C., Natraj, V., Neil, D., Nowlan, C., O'Sullivan, E., Palmer, P., Pierce, R., Pippin, M., Saiz-Lopez, A., Spurr, R., Szykman, J., Torres, O., Veefkind, J., Veihelmann, B., Wang, H., Wang, J., and Chance, K.: Tropospheric emissions: Monitoring of pollution (TEMPO), Journal of Quantitative Spectroscopy and Radiative Transfer, 186, 17–39, https://doi.org/https://doi.org/10.1016/j.jqsrt.2016.05.008, 2017.

---

## Author Comment (AC2)

We would like to thank the referee for the useful comments and constructive suggestions. In the following, we address the referee's comments and describe corresponding changes we have made to the manuscript. The referee's comments are listed in *italics*, followed by our response in blue. New/modified text in the manuscript is in **bold**.

*Getting the best estimate of the PBLH is a worthwhile endeavor, which can help with our basic understanding of processes associated with the PBL and can be used to help modelling. The manuscript is suitably organized, is generally well written, and includes appropriate figures. As with any empirical technique, the interpretation needs to be done in a way that clearly points out the limitations and bias of choices. In addition, the case must be made that these results are robust. In particular, are the results actually generalizable or is it too dependent on what is or is not included in both the method (e.g., parameter selection) and training data (e.g. which airports or years used). This is a common issue in applying any sort of empirical method, but it is really important to be clear about this. Otherwise, one could repeat these steps, but alter a couple of choices, and come up with different results.*

We thank the referee for this important comment. In the revised manuscript, we point out limitations of our study and provide all the key details of our model such as parameters and training data (see our responses to several comments below). We revise the sentences at lines 69-75 to the following:

**"The existing AMDAR PBLH dataset is available hourly from 2005 to 2019 at 54 airport locations. However, many applications require PBLH in other locations or completely covering a region. The objective of this study is to produce observation-based, spatially-complete PBLH fields over the CONUS. We develop a data-driven predictive model using various meteorological and geographical predictors to match the AMDAR PBLH observations. Since the predictors all have complete spatial coverage over the CONUS, running the model forwardly can yield PBLH prediction at arbitrary locations in the domain. We cross-validate the model in space by randomly splitting the airports into training and testing sets, and the model is selected based on averaged metrics on the testing sets. The predicted PBLH is then compared to PBLHs from three widely-used reanalysis products (ERA5, MERRA-2, and NARR), and all of them are further compared to independently diagnosed PBLH from observations (e.g., from research aircraft profiles and HSRL airborne lidar) and the CALIPSO PBLH product."**

We also discuss the generalizability of our work by adding the following sentence to the first paragraph of the conclusions:

**"The model configurations have been specifically optimized and evaluated to generate the spatially complete PBLH dataset over the CONUS during the period when AMDAR PBLH data are available (2005–2019). Further generalization of the work will require additional tuning and model evaluation."**

*Input data: Data is preferentially excluded. For instance, if the AMDAR PBLH is too far away from the ERA5 estimate it is thrown out, so as stated around line 115, "...which accounted for about one third of all AMDAR data, half of data under stable condition, and only 10% of data under convective condition." This has many implications for all subsequent analysis. Importance of the ERA5 PBLH (at times 0, -1, and -2) for permutation and SHAP feature is extremely large (Fig. 5). Throwing out all "bad" AMDAR data contributes to that importance, and basically implies overfitting to the ERA5 PBLH. The method itself accounts for overfitting, but if the input data is already filtered to get rid of 'bad' data before the method is applied, it will artificially create a 'better' fit.*

Thanks for this important comment. It is not uncommon to detect and correct corrupt or inaccurate records from the dataset before any valid statistical model can be built. In this case, the data cleaning is a crucial preprocessing step. As seen from Fig. 2a-b in the manuscript, a portion of data points are characterized by very low ERA5 PBLH but much higher AMDAR PBLH. We have tried to explain this "troubled group" using a wide range of variables but did not find any meaningful correlation. The most likely cause of the troubled group is the ambiguity of PBL top diagnosed from ERA5 and AMDAR. Although both ERA5 and AMDAR use the critical bulk Richardson number method, the footprint of their profiles is very different. ERA5 profiles supposedly represent averages over a 0.25° grid cell and AMDAR profiles (at least below a few km) are in-situ measurements at points.

These challenging cases happen much more often in stable and neutral conditions, suggesting that they are likely due to ERA5 and AMDAR identifying different vertical structures as the PBL top. Including this troubled group will derail the whole model fitting as the algorithm would have struggled to explain the large differences between AMDAR and ERA5 PBLH, which is the dominant feature. To clarify this point, we revise the sentences at lines 106–108 to the following:

**"We did not find any meteorological or geographical factors that would explain the occurrence of AMDAR vs. ERA5 PBLH data pairs in these clusters. The most likely cause of these clusters of data is the ambiguity of identifying the PBL top. Under challenging conditions, the critical bulk Richardson number algorithm that is used in AMDAR and ERA5 data to diagnose PBLH may identify different vertical structures as PBL top, leading to very different and uncorrelated PBLH values. Including these uncorrelated clusters of points will strongly bias the model training results."**

We do understand the referee's concern and to further address the this concern, we explicitly acknowledge this data preprocessing step in the conclusions as a limitation of our study after line 337:

**"Significant challenges still exist due to the lack of PBLH observations and the uncertainty of existing datasets. We observe clusters of AMDAR observations that are uncorrelated with collocated ERA5 PBLH, mostly under stable conditions, and no meteorological or geographical factors could explain this discrep-**

**ancy. A preprocessing step had to be implemented to mitigate its impacts on the model training. The satellite-based CALIPSO dataset is the most spatiotemporally complete for model evaluation, but it is subject to large uncertainties, which gives essentially no correlations with reanalysis datasets and the prediction from this work when PBLH is lower than 1 km. Future spaceborne PBLH observations with higher fidelity and more routine suborbital measurements, especially under stable conditions, will be beneficial."**

*Comparisons: Fig. 3 gives distributions of PBLH from various datasets at various locations, times, and sample sizes. If the point of the figure is to show how different places and times have different distributions of PBLH, is this really necessary? If the point of this figure is to compare distributions of PBLH obtained from different data sets, then the data sets must use the same locations and times for a fair comparison. Otherwise, the differences seen in the plot have no meaning since the differences could just be a result of when it was sampled. As it is now in Fig. 3a, CALIPSO and AMDAR have extremely different distributions, so the results of essentially no relationship in Fig. 7 is not surprising.*

We deem that the point of this figure is neither "to show how different places and times have different distributions of PBLH" nor "to compare distributions of PBLH obtained from different data set". We would like to think of this figure as a necessary overview of PBLH from various datasets, which sets the stage for the following sections.

We completely understand the reviewer's concern about comparison with CALIPSO. Besides a few more revision to be brought up in the following responses, we add a note that the distributions of CALIPSO and AMDAR should not be directly compared given their different sampling. The sentence at lines 157–158 is revised to:

**"One should note that the distributions of CALIPSO and AMDAR should not be directly compared given their different sampling and the large uncertainty from CALIPSO."**

Nonetheless, one could compare the spirals (in-situ profiles) and HSRL (airborne lidar), which did happen during the same campaigns, in Fig. 3.

*Mountain West: Given the high average PBLH in the mountain west compared to the rest of the country, the variance is likely to be much larger as well. This has a couple major implications. First, any differences between data sets are likely dominated by differences in the mountain west. Has this been assessed with this data set? This could be done fairly easily in two ways. Either use only the eastern or western half of CONUS and repeat the analysis, or normalize by PBLH. Again, because this is an empirical method, the results could be much different by sector.*

Figure 2 in the manuscript does not seem to suggest that high PBLH values are associated with high variances. Moreover, the RMSEs of XGB vs. HSRL and spiral (Fig. 9c and Fig. 10c) do not show outstandingly high variances in Colorado than other regions. We have tested normalization by fitting the log of PBLH early on in this study, but the results

were not as good as using the PBLH. In the revised manuscript, we do acknowledge that a further step of our work can be separating the CONUS into different geographic regions, as the reviewer suggested. The following is added after line 344:

**"Future improvements of model performance may be achieved by focusing on smaller geographic regions and fine tuning region-specific predictors."**

*PBLH Reference: With PBLH, as we are all aware, there is no 'gold standard' that is a reliable reference for comparison given limitations in spatial or temporal resolution, retrieval method, etc. When comparing XGB with the reanalysis and CALIOP, it is not clear if the same time periods are used. For instance, AMDAR used 2005 to 2019 AMDAR (Line 189), but CALIOP from 2006 to 2013 (Line 150). So do all these comparisons use a consistent period of time? If not, this may lead to biases from using different times.*

After the XGB model is trained using AMDAR data (from 2005 to 2019 as the reviewer correctly pointed out), it can be used to predict PBLH at any other locations and times within the domain, so the comparisons in sections 4.1-4.3 are consistent in space and time. The following sentence is added to line 151 in the original manuscript to clarify this point:

**"During the evaluation, we first obtain the model prediction at the same location and time of each CALIPSO sounding and then compare the predicted PBLH with the CALIPSO PBLH."**

For a more thorough discussion about the PBLH references, we group the descriptions of the three evaluation datasets (CALIPSO, HSRL, and spirals) into a single subsection ("**2.3 Observational datasets used for evaluation**") and add a new subsubsection to overview the pros and cons of these datasets:

**"2.3.4 Comparisons of observational datasets**

**As summarized by Figs. 1 and 3, none of the observational datasets described above can uniformly represent the PBLH over the study domain. CALIPSO features the most homogeneous spatial coverage (Fig. 1b), but its PBLH product relies on an automatic, global algorithm that may be subject to significant uncertainties. Yet the unique benefit of including CALIPSO data is that it can indicate errors in the spatial prediction made by our model, as the availability of AMDAR airports is spatially clustered (Fig. 1a). For example, no AMDAR sites are available in the large area over the Northern Rockies and Plains and the Southeast. Because of the large differences in AMDAR and CALIPSO PBLH, we consider the intercomparison involving CALIPSO more relative than absolute and focus on correlations rather than biases.**

**One should also note that CALIPSO and HSRL PBLH data are based on aerosol backscatter gradients, which is quite distinct from AMDAR, DISCOVER-AQ spiral profiles, and ERA5, where PBLH values are diagnosed thermodynamically. Although systematic differences between aerosol-based and thermodynamics-based PBLH may exist, we do not observe them by comparing spatiotemporally**

**close spiral and HSRL measurements in the same DISCOVER-AQ campaigns (i.e., comparing d vs. g, c vs. h, d vs. i, and e vs. j in Fig. 3). Furthermore, the model prediction from this work may serve as a "traveling standard" when evaluated against HSRL and spiral datasets. As will be shown in Sections 4.2 and 4.3, the biases between HSRL data and collocated model prediction do not show significant differences from the biases between spiral data and the corresponding model prediction."**

*Tuning and Training: Selecting 800 trees with a depth of 8, which is a large amount, still results in a rather large IQR for the test set, even considering differences of sample size. If this were just an issue with large variance, at least the IQRs would overlap. None of the IQRs between training and testing overlap (and even the 97.5 percentiles barely overlap!), suggesting little utility of using this method outside of the training data. This really points to some large underlying flaw, which could be related to a number of factors.*

The number of trees, the tree depth, and a few other hyperparameters were determined from the data by cross validation using various metrics on the testing dataset (Section 3.2). The selected hyperparameters are the best performers on the testing data on average. Since the main motivation of this study is to fill the spatial gaps between AMDAR sites, we put more weight on the model performance than the simplicity of model. The model predictions are evaluated on three other observational datasets in Section 4, and neither large biases nor large variances are observed on locations away from AMDAR sites. The following sentence is added to line 220 of the original manuscript:

**"Although driven by the data, this selection yields a complicated XGB model. Since the main motivation of this study is to fill the spatial gaps between AMDAR sites over the CONUS during the AMDAR period without further extrapolating in space or time, we put more weight on the model performance than the simplicity or computational cost of model."**

It is unclear to us why *"None of the IQRs between training and testing overlap"* would suggest *"little utility of using this method outside of the training data"*. If there were really little utility of using the method outside of training set, the metrics on testing should approach a null model, i.e., zero $R^2$ and RMSE approaching the variance of the predicted variable. That's not what we observe in Fig. 4 of the manuscript.

*Line 125: A good reason to use AMDAR and ERA5 is that they can both use the bulk Richardson number to find PBLH. Even though a critical Ri of 0.5 was used in a previous study with AMDAR, why shouldn't this work use a consistent critical Ri?*

The AMDAR and ERA5 profiles are fundamentally different as one is measured in-situ, while the other is model-based and on a $0.25°$ grid. AMDAR profiles contain structures that cannot be resolved by ERA5. Hence the optimal parameters for both profiles are unlikely the same. In addition to a critical Ri of 0.25, ERA5 also used another parameter $b = 0$, whereas AMDAR used $b = 100$ (Zhang et al., 2020). Zhang et al. (2020) compared AMDAR

PBLH estimated with the same parameters as ERA5, but the results gave larger differences from ERA5 and larger biases relative to other observation datasets. We revised the sentence at line 125 accordingly:

**"The PBLH from the ERA5 product is identified using the bulk Richardson number method but with slightly different parameters (ECMWF, 2017). Zhang et al. (2020) compared AMDAR PBLH estimated with the same parameters as ERA5, but the results gave larger biases relative to ERA5 and other observations."**

*Line 252: Using year as a factor in the final model is a surprising feature since there is no physical basis for this. This suggests that if extending to a new year of 2022, it is not possible to use relationships developed in this model, so it calls the generality or robustness of the model into question.*

The year as predictor can capture any interannual variation that cannot be explained by the features. Similarly we tested day of year for any remaining potential seasonal variation, but day of year is ranked low in the feature important tests and not included. See lines 236–238 of the original manuscript. The main motivation is to fill the spatial gap when the AMDAR data are available, so we will not simply extrapolate to future years (revisions have been made in the previous responses). However, when the AMDAR data in 2022 become available, the model can be developed in the same way. We add the following sentence to address this point:

**"The significance of year as a predictor indicates interannual variations that cannot be explained by other physics-based predictors. Therefore, this model should be used during the years when AMDAR data are available."**

*Fig. 5: Because the BL height at time 0, -1, and -2 is so important in this model, do you think that a linear trend would work just as well to get the BL height? If so, the simpler model is better.*

We tested linear model using the same predictors, the RMSE on testing sets are about 5% higher than XGB and 3% higher than random forest. We add the following to line 204:

**"The linear regression model, although the simplest and fastest, was not used because of its slightly lower performance than XGB (testing RMSE higher by $\sim 5\%$) and random forest (testing RMSE higher by $\sim 3\%$) and the fact that the computing cost of XGB is not of concern."**

*Section 4.1: Using CALIPSO as a benchmark seems problematic; there are many issues with PBLH retrievals from CALIPSO, and Fig. 7 shows that there is really no agreement at all with any data set to CALIPSO.*

We thank the referee for this comment, which was also raised by the other referee. We were not necessarily treating CALIPSO as a benchmark, but an independent, observation-based sanity check at locations without AMDAR sites (e.g., Fig. 8 shows no excessive biases at

locations without AMDAR sites). This was perhaps not clear in the previous submission but is now clarified in multiple places in the revised manuscript, including abstract, introduction, and the newly added section 2.3.4.

*Line 340: Yes, a natural next step is to extend it to other times, but the above issues would be much worse given the added difficulty of defining the nocturnal boundary layer.*

Yes, we completely agree with the referee on the difficulty in defining the height of nocturnal boundary layer, which is exactly why we need future studies on this topic. We revise this sentence to:

**"Since the AMDAR PBLH data are available hourly, it is possible to extend this work to other daytime hours and even nighttime hours with the caution that it will be more challenging due to the increase of stable conditions and less observational datasets available for evaluation."**

**References**

ECMWF: Part IV: Physical processes, in: IFS Documentation CY43R3, IFS Documentation, ECMWF, URL `https://www.ecmwf.int/node/17736`, 2017.

Zhang, Y., Sun, K., Gao, Z., Pan, Z., Shook, M. A., and Li, D.: Diurnal Climatology of Planetary Boundary Layer Height Over the Contiguous United States Derived From AMDAR and Reanalysis Data, Journal of Geophysical Research: Atmospheres, 125, e2020JD032 803, https://doi.org/https://doi.org/10.1029/2020JD032803, e2020JD032803 2020JD032803, 2020.

---

## Author Response (AR2)

**Responses to referee 1:**

We would like to thank the referee for the useful comments and constructive suggestions. In the following, we address the referee's comments and describe corresponding changes we have made to the manuscript. The referee's comments are listed in *italics*, followed by our response in blue. New/modified text in the manuscript is in **bold**.

*The authors have addressed most of my concerns, and the effort they put into the response is appreciated. Just a couple things need to be addressed, which may not have described as well as necessary in the initial review.*

*Figure 4 is still troubling me. Normally when the testing errors are higher than the training errors and there is no overlap at all between training and testing errors, this suggests overfitting. Overfitting is a modeling error that introduces bias into the model because the model is too closely related to the training data set. In fact, it is so closely related that it captures the noise in the training dataset and becomes less relevant to any other data set (i.e., the testing set, or eventually to using the method to interpolate between locations with AMDAR data). If the evolution of training and testing errors are examined, the training error will always decrease with more trees, depth, etc. The testing error will also decrease, but the testing error will eventually reach a minimum and then increase again. If the number of trees, depth, and other hyperparameters suggest that this is really the optimal configuration and not overfitted, then there could issues coming from something else. In any event, please include the number of leaves that are being used.*

We thank the referee for this important insight. We set the max number of leaves to the default, which does not limit the number of leaves. We acknowledge this limitation by revising the sentence at lines 266–267 of the manuscript:

**"We note that the interquartile ranges of the metrics' distributions on the training and testing datasets do not overlap. This indicates a certain level of overfitting still exists, and the model may be further improved by tuning more hyperparameters other than the number of trees and the maximum tree depth. For example, the max number of leaves is at default value which is unlimited."**

*The last concern is again with throwing out the uncorrelated AMDAR-ERA5 data pairs. It is clear that including these will add uncertainty to the model, so the model is developed without these uncorrelated pairs. However, if the intent is to then use this model to fill the spatial gaps where there is no AMDAR data, it is not possible to throw out pairs that would be uncorrelated since there is no AMDAR data to perform the initial step of removing those uncorrelated data pairs. So, a model would be applied to a subset of data that is not accounted for in the initial construction of the model. These issues should at least be clearly stated so a reader is aware of the limitations.*

When we use the trained model to make predictions, only the predictors will be needed, so the fact that *"it is not possible to throw out pairs that would be uncorrelated since there is no AMDAR data to perform the initial step of removing those uncorrelated data pairs"*

does not cause a technical problem. We do acknowledge that the model is constructed on a subset of AMDAR data, whereas when making predictions, all conditions will be included. The sentence at line 391 is revised to:

**"A preprocessing step that filters out these data clusters had to be implemented to mitigate its impacts on the model training. Consequently, the model is trained and tested on a subset of AMDAR data and does not fully represent the entire AMDAR dataset."**

This limitation is also acknowledged in the abstract by adding the following to line 4 of the manuscript:

**"A preprocessing step was implemented to exclude AMDAR data points that were unexplainable by the predictors, mostly under stable conditions."**